# Dueling Bandits with Adversarial Sleeping

**Aadirupa Saha**[*]      Pierre Gaillard [†]

## Abstract

We introduce the problem of sleeping dueling bandits with stochastic preferences and adversarial availabilities (DB-SPAA). In almost all dueling bandit applications, the decision space often changes over time; eg, retail store management, online shopping, restaurant recommendation, search engine optimization, etc. Surprisingly, this 'sleeping aspect' of dueling bandits has never been studied in the literature. Like dueling bandits, the goal is to compete with the best arm by sequentially querying the preference feedback of item pairs. The non-triviality however results due to the non-stationary item spaces that allow any arbitrary subsets items to go unavailable every round. The goal is to find an optimal 'no-regret' policy that can identify the best available item at each round, as opposed to the standard 'fixed best-arm regret objective' of dueling bandits. We first derive an instance-specific lower bound for DB-SPAA $\Omega(\sum_{i=1}^{K-1}\sum_{j=i+1}^{K}\frac{\log T}{\Delta(i,j)})$, where $K$ is the number of items and $\Delta(i,j)$ is the gap between items $i$ and $j$. This indicates that the sleeping problem with preference feedback is inherently more difficult than that for classical multi-armed bandits (MAB). We then propose two algorithms, with near optimal regret guarantees. Our results are corroborated empirically.

## 1 Introduction

The problem of *Dueling-Bandits* has gained much attention in the machine learning community [34, 38, 36], which is an online learning framework that generalizes the standard multiarmed bandit (MAB) [6] setting for identifying a set of 'good' arms from a fixed decision-space (set of arms/items) by querying preference feedback of actively chosen item-pairs. More formally, in dueling bandits, the learning proceeds in rounds: At each round, the learner selects a pair of arms and observes stochastic preference feedback of the winner of the comparison (duel) between the selected arms; the objective of the learner is to minimize the regret with respect to a (or set of) 'best' arm(s) in hindsight. Towards this several algorithms have been proposed [2, 37, 21, 14]. Due to the inherent exploration-vs-exploitation tradeoff of the learning framework and several advantages of preference feedback [9, 35], many real-world applications can be modeled as dueling bandits, including movie recommendations, retail management, search engine optimization, job scheduling, etc.

However, in reality, the decision spaces might often change over time due to the non-availability of some items, which are considered to be 'sleeping'. This 'sleeping-aspect' of online decision making problems has been widely studied in the standard multiarmed bandit (MAB) literature [17, 24, 15, 19, 18, 11]. There the goal is to learn a 'no-regret' policy that maps to the 'best awake item' of any available (non-sleeping) subset of items, and the learner's performance is measured with respect to the optimal policy in hindsight. This setting is famously known as *Sleeping Bandits* in MAB [17, 24, 15, 11]. More discussions are given in Related Works.

Surprisingly, however, the *'sleeping problem'* is completely unaddressed in the preference bandits literature, even for the special case of pairwise preference feedback, which is famously studied

---

[*]Microsoft Research, New York, US; `aasa@microsoft.com`.
[†]Univ. Grenoble Alpes, Inria, CNRS, Grenoble INP, LJK, 38000 Grenoble, France. `pierre.gaillard@inria.fr`

35th Conference on Neural Information Processing Systems (NeurIPS 2021).

---

**Parameters.** Item set: $[K]$ (known), Preference: $\mathbf{P}$ (unknown), Available item sets: $\mathcal{S}_T$ (observed sequentially)

**For** $t = 1, 2, \ldots, T$, the learner:
- Observes $S_t \subseteq [K]$ the set of available items
- Chooses $(x_t, y_t) \in S_t^2$
- Observes $o_t := \mathbf{1}(x_t \succ y_t) \sim \text{Ber}(\mathbf{P}(x_t, y_t))$
- Incurs $r_t := \frac{1}{2}\big(\mathbf{P}(i_t^*, x_t) + \mathbf{P}(i_t^*, y_t) - 1\big)$;
   where $i_t^*$ is such that $\min_{j \in S_t} \mathbf{P}(i_t^*, j) \geq \frac{1}{2}$

---

Figure 1: Setting of DB-SPAA($\mathbf{P}, \mathcal{S}_T$)

as *Dueling Bandits* [37, 34], even though the setup of changing decision spaces are quite relevant in almost every practical applications: Be that in retail stores where some items might go out of production over time, for search engine optimization some websites could be down on certain days, in recommender systems some restaurants might be closed or movies could be outdated, in clinical trials certain drugs could be out of stock, and many more. This work is the first to consider the problem of *Sleeping Dueling Bandits*, where we formulated the stochastic $K$-armed dueling bandit problem with adversarial item availabilities. Here at each round $t \in \{1, 2, \ldots, T\}$ the item preferences are considered to be generated from a fixed underlying (and of course unknown) preference matrix $\mathbf{P} \in [0, 1]^{K \times K}$, however, the set of available actions $S_t \subseteq \{1, 2, \ldots, K\}$ is assumed to be adversarially chosen by the environment. We call the problem as *Sleeping-Dueling Bandit with Stochastic Preferences and Adversarial Availabilities* or in brief DB-SPAA($\mathbf{P}, \mathcal{S}_T$), where $\mathcal{S}_T = \{S_1, S_2, \ldots S_T\}$ denotes the sequence of available subsets over $T$ rounds. We also assume the preference $\mathbf{P}$ follows a 'total-ordering assumption to ensure the existence of a best-item per available subset $S_t$. We describe the setting in Fig. 1 with a formal description in Sec. 2.

Our specific contributions are as follows:

**1.** We first analyze the fundamental performance limit for the DB-SPAA($\mathbf{P}, \mathcal{S}_T$) problem in Sec. 3: Thm. 1 gives an instance-specific regret lower bound of

$$\Omega\big( \sum_{i=1}^{K-1} \sum_{j=i+1}^{K} \frac{\log T}{\Delta(i,j)} \big),$$

with $\Delta(i, j)$ being the 'preference gap' of item $i$-vs-$j$ (see Eqn. (2)). Our lower bound, which can be of order $\Omega(K^2 \log T / \Delta)$, $\Delta := \min_{i,j} \Delta(i, j)$ being the worst case gap, indicates that the *problem of sleeping dueling bandits is inherently more difficult that standard sleeping bandits (MAB)*, unlike the 'non-sleeping' case where both dueling bandits (with 'total-ordering' assumption on $\mathbf{P}$) and MAB are known to have the same fundamental performance limit of $\Omega(K \log T / \Delta)$ (Rem. 1).

**2.** We next design a *'fixed confidence regret'* algorithm SlDB-UCB (Alg. 1), inspired from the pairwise upper confidence bound (UCB) based algorithm [37]. However due to the fixed confidence and 'adversarial-sleeping' nature of the problem, we need to differently maintain pairwise confidence bounds per item (based on the availability sequence $\{S_t\}$), which makes the resulting algorithm and its subsequent analysis significantly different than standard UCB based dueling bandit algorithms: Precisely given any $\delta > 0$, SlDB-UCB achieves a regret of $O\Big( \frac{K^3 \log(1/\delta)}{\Delta^2} \Big)$ with probability at least $1 - \delta$ over any problem instance of DB-SPAA($\mathbf{P}, \mathcal{S}_T$) (Sec. 4).

**3.** In Sec. 5, we design another computationally efficient algorithm, SlDB-ED (Alg. 2), for *'expected regret'* guarantee. Unlike the previous algorithm (SlDB-UCB), SlDB-ED uses empirical divergence (ED) based measures to filter out the 'good' set of arms, inspired from the idea of RMED algorithm of [21] for standard dueling bandits; however, due to sleeping nature of the items, it requires a different maintenance of 'good' arms and the regret analysis of the algorithm requires derivation of new results (as described in Sec. 5). The algorithm is shown to perform near optimally with an expected *non-asymptotic* regret upper-bound of (Thm. 6). Note that for any problem instance with constant suboptimality gaps $\Delta(i, j) = \Delta$ for all $i < j$, regret bound of SlDB-ED is tight and matches the lower-bound ensuring the near optimality of SlDB-ED in the worst case. Furthermore, a novelty of our finite time regret analysis lies in showing a cleaner tradeoff between regret vs. availability sequence $\mathcal{S}_T$ which automatically adapts to the inherent 'hardness' of the sequence of available

subsets $\mathcal{S}_T$, compared to existing sleeping bandits work for adversarial availabilities in the MAB setting [19] which only gives a worst-case regret bound over all possible availability sequences (Rem. 3).

**4.** Finally we corroborate our theoretical results with extensive empirical evaluations. (Sec. 6).

**Related Works.** The problem of regret minimization for stochastic multiarmed bandits (MAB) is extremely well studied in the online learning literature [6, 1, 22, 5, 16], where the learner gets to see a noisy draw of absolute reward feedback of an arm upon playing a single arm per round.

A well motivated generalization of MAB framework is *Sleeping Bandits* [17, 24, 18, 15], much studied in the online learning community, where at any round the set of available actions could vary stochastically based on some unknown distributions over the decision space of $K$ items [24, 11] or adversarially [15, 19, 18]. Besides the reward model, the set of available actions could also vary stochastically or adversarially [17, 24]. The problem is NP-hard when both rewards and availabilities are adversarial [19, 18, 15]. In case of stochastic reward and adversarial availabilities [19] proposed an UCB based no-regret algorithm, which was also shown to be provably optimal. The case of adversarial reward and stochastic availabilities has also been studied where the achievable regret lower bound is known be $\Omega(\sqrt{KT})$ by the inefficient EXP4 algorithm [19, 15].

On the other hand over the last decade, the relative feedback variants of stochastic MAB problem has seen a widespread resurgence in the form of the Dueling Bandit problem, where, instead of getting noisy feedback of the reward of the chosen arm, the learner only gets to see a noisy feedback on the pairwise preference of two arms selected by the learner. The objective of the learner is to minimize the regret with respect to 'best arm in the stochastic model. Several algorithms have been proposed to address this dueling bandits problem, for different notions of 'best arms' or preference models [10, 31, 38, 37, 36, 21, 33, 13], or even extending the pairwise preference to subsetwise preferences [29, 8, 26, 27, 25]. However, surprisingly, unlike the 'sleeping bandits generalization' of MAB, no parallel has been drawn for dueling bandits, which remains our main focus.

## 2 Problem Formulation

**Notations.** Decision space (or item/arm set) $[K] := \{1, 2, \ldots, K\}$. The available set of items at round $t$ is denoted by $S_t \subseteq [K]$. For any matrix $\mathbf{M} \in \mathbb{R}^{K \times K}$, we define $m_{ij} := M(i, j)$, $\forall i, j \in [K]$. We write $S_{\backslash i} = S \setminus \{i\}$, for any $S \subseteq [K]$ and $i \in S$. $\mathbf{1}(\cdot)$ denotes the indicator random variable which takes value $1$ if the predicate is true and $0$ otherwise and $\lesssim$ a rough inequality which holds up to universal constants. For any two items $x, y \in [K]$, we use the symbol $x \succ y$ to denote $x$ is preferred over $\mathbf{y}$. $\mathbf{\Sigma}_K$ denotes the set of all permutations of the items in set $[K]$. The KL-divergence of two Bernoullis with biases $p$ and $q$ respectively is written $\mathrm{kl}(p, q) := p \log(p/q) + (1 - p) \log((1-p)/(1-q))$. We assume $\frac{0}{0} := 0.5$ (in Alg. 1 and 2).

**Setup.** We consider the problem of stochastic $K$-armed dueling bandits with adversarial availabilities: At every iteration $t = 1, \ldots, T$, a set of available items (actions) $S_t \subseteq [K]$ is revealed, and the learner is asked to choose two items $x_t, y_t \in S_t$. Then, the learner receives a preference feedback $o_t = \mathbf{1}(x_t \succ y_t) \sim \mathrm{Ber}(\mathbf{P}(x_t, y_t))$, where $\mathbf{P} \in [0, 1]^{K \times K}$ is an underlying pairwise preference matrix, unknown to the learner. The setting is described in Figure 1. We assume that $\mathbf{P}$ respects a *'total ordering'*, say $\sigma^* \in \mathbf{\Sigma}_K$. Without loss of generality, we set $\sigma^* = (1, 2, \ldots, K)$ thoughout the paper. This implies $\mathbf{P}(i, j) \geq 0.5$ for $i \leq j$. One possible pairwise probability model which respects *'total ordering'* is Plackett-Luce [7], where it is assumed that the $K$ items are associated to positive score parameters $\theta_1, \ldots, \theta_K$, and $\mathbf{P}(i, j) = \theta_i / (\theta_i + \theta_j)$ for all $i, j \in [K]$. In fact any well random utility (RUM) based preference model would have the above property, like [7, 28]. Note also that our assumption corresponds to assuming the existence of a Condorcet winner for every subset $S_t \subseteq [K]$.

**Objective.** The objective of the learner is to minimize his regret over $T$ rounds with respect to the best policy in the policy class $\Pi = \{\pi : 2^K \mapsto [K] \mid \forall t \in [T], \pi(S_t) \in S_t\}$, i.e. any $\pi \in \Pi$ is such that for any $t \in [T]$, $\pi(S_t) \in S_t$. More formally we define the regret as follows:

$$R_T = \max_{\pi \in \Pi} \sum_{t=1}^{T} \frac{\mathbf{P}(\pi(S_t), x_t) + \mathbf{P}(\pi(S_t), y_t) - 1}{2}. \tag{1}$$

We analyze both *fixed-confidence* and *expected* regret guarantees in this paper respectively in Sec. 4 (see Thm. 3) and Sec. 5 (see Thm. 6). It is easy to note that under our preference modelling assumptions, the best policy, say $\pi^*$, turns out to be $\pi^*(S) = \min\{S\}$ for any $S \subseteq [K]$. We henceforth denote by $i_t^* = \pi^*(S_t)$. We define the above problem to be *Sleeping-Dueling Bandit with Stochastic Preferences and Adversarial Availabilities* over the stochastic preference matrix $\mathbf{P} \in [0,1]^{K \times K}$ and the sequence of available subsets $\mathcal{S}_T = \{S_1, \ldots, S_T\}$, or in short DB-SPAA($\mathbf{P}, \mathcal{S}_T$). For ease of notation we respectively define the gaps and the non-zeros gaps as $\Delta(i,j) := \mathbf{P}(i,j) - 1/2$, and

$$\Delta(i,j)_+ := \begin{cases} \Delta(i,j) & \text{if } \Delta(i,j) \neq 0 \\ +\infty & \text{if } \Delta(i,j) = 0 \end{cases} \tag{2}$$

The regret thus can be rewritten as $R_T := \sum_{t=1}^{T} r_t$, where $r_t := (\Delta(i_t^*, x_t) + \Delta(i_t^*, y_t))/2$ denotes the instantaneous regret. We also denote by $n_{ij}(t) := \sum_{\tau=1}^{t} \mathbf{1}(\{x_t, y_t\} = \{i,j\})$ the number of times the pair $(i,j)$ is played until time $t$ and by $w_{ij}(t)$ the number of times $i$ beats $j$ in $t$ rounds.

## 3 Lower Bound

We first derive a worst case regret lower bound over all possible sequences of $\mathcal{S}_T$. The proof idea essentially lies in constructing *hard enough* availability sequences $\mathcal{S}_T$, where no learner can escape learning the preferences of every distinct pair of items $(i,j)$. This leads to a potential lower bound of $\Omega\big(K^2 \log(T)/\Delta\big)$. For this section we denote $\mathbf{P}$ by $\mathbf{P}_K$ to make the dependency on $K$ more precise.

**Theorem 1** (Lower Bound for DB-SPAA($\mathbf{P}_K, \mathcal{S}_T$)). *For any No-regret learning algorithm $\mathcal{A}$, there exists a problem instance DB-SPAA($\mathbf{P}_K, \mathcal{S}_T$) with $T \geq K^4$, such that its expected regret is lower-bounded as:*

$$\mathbf{E}[R_T(\mathcal{A})] \geq \Omega\bigg( \sum_{i=1}^{K-1} \sum_{j=i+1}^{K} \frac{\log T}{\Delta(i,j)_+} \bigg).$$

The *No-regret learning algorithm* refers to the class of 'consistent algorithms' which do not pull any suboptimal pair more than $O(T^\alpha)$, $\alpha \in [0,1]$ (see Def. 7, Appendix 8)).

**Proof (sketch)** The main argument lies behind the fact that in the worst case the adversary can force the algorithm to learn the preference of every distinct pair $(i,j)$ as for the 'worst-case' sequences $\mathcal{S}_T$, a knowledge of the already 'learnt' pairwise preferences would not disclose any information on the remaining pairs; e.g. assuming $\sigma^* = (1, 2, \ldots, K)$, revealing the available subsets in the following sequence $(1, 2), (1, 3), \ldots (1, K), (2, 3), (2, 4), \ldots (K-1, K)$ would force the learner to explore (learn the preferences) all $\binom{K}{2}$ distinct pairs. The remaining proof establishes this formally, towards which we first show a $\Omega(\ln(T)/\Delta(1,2))$ regret lower bound for a DB-SPAA instance with just two items (i.e. $K = 2$) as shown in Lem. 2. The lower bound for any general $K$ can now be derived applying the above bound on independent $\binom{K}{2}$ subintervals, with the availability sequence $(1, 2), (1, 3), \ldots (1, K), (2, 3), (2, 4), \ldots (K-1, K)$. The full proof is given in Appendix 8. $\square$

**Lemma 2** (Lower Bound of DB-SPAA($\mathbf{P}_K, \mathcal{S}_T$) for 2 items). *For any No-regret learning algorithm $\mathcal{A}$, there exists a problem instance DB-SPAA($\mathbf{P}_2, \mathcal{S}_T$) such that the expected regret incurred by $\mathcal{A}$ on that can be lower bounded as: $\mathbf{E}[R_T(\mathcal{A})] \geq \Delta^{-1} \log(T)$, $\Delta$ being the 'preference-gap' between the two items (i.e. $\Delta = \mathbf{P}_{12} - 1/2$, assuming $P_{12} > 1/2$ or equivalently $\Delta > 0$).*

**Remark 1** (Implication of the lower bound). The above result indicates that in the preference based learning setup, the fundamental problem complexity lies in distinguishing every pair of items $1 \leq i < j \leq K$. If the learner fails to learn the preference of any pair $(i,j)$, the adversary can make the learner suffer $O(T)$ regret by setting $S_t = \{i,j\}$ henceforth at all round. It is worth noting that in the 'no sleeping' case both dueling bandits and MAB are known to have the same fundamental performance limit of $\Omega(K \log(T)/\Delta)$ (assuming $\mathbf{P}$ respects a *condorcet winner* [6, 21]). Thus Thm. 1 shows that the 'sleeping-aspect' of dueling bandits makes the problem $K$-times harder than 'sleeping-MAB' for which the regret lower bound is known to be only $\Omega\big(\sum_{i=1}^{K} \log(T)/\Delta(i, i+1)\big)$ [19].

---
**Algorithm 1** `SlDB-UCB`
---

1: **input:** Arm set: $[K]$, parameters $\alpha > 0.5$, Confidence parameter $\delta \in [0, 1)$
2: **init:** $w_{ij}(1) \leftarrow 0, \forall i, j \in [K]$.
3: **define:** $n_{ij}(t) := w_{ij}(t) + w_{ji}(t), \forall t \in [T]$
4: **for** $t = 1, 2, \ldots, T$ **do**
5:     Receive $S_t \subseteq [K]$
6:     $\widehat{p}_{ij}(t) = \dfrac{w_{ij}(t)}{n_{ij}(t)}$, $c_{ij}(t) \leftarrow \sqrt{\dfrac{\alpha \log a_{ij}(t)}{n_{ij}(t)}}$, $\forall i, j \in S_t$, (assume $\frac{x}{0} := 0.5, \ \forall x \in \mathbb{R}$)
7:     $u_{ij}(t) \leftarrow \widehat{p}_{ij}(t) + c_{ij}(t)$, $u_{ii}(t) \leftarrow \frac{1}{2}$, $\forall i, j \in S_t$        ▷ UCB of empirical preferences
8:     **for** $k \in S_t$ **do**
9:        $\mathcal{C}_k(t) := \{j \in S_t \mid u_{kj}(t) > \frac{1}{2}\}$        ▷ Potential losers to $k$
10:     **end for**
11:     $\mathcal{C}_t = \{i \in S_t \mid |\mathcal{C}_i(t)| = \max_{j \in S_t} |\mathcal{C}_j(t)|\}$        ▷ Potential best items
12:     Select a random $x_t$ from $\mathcal{C}_t$. Choose $y_t \leftarrow \arg\max_{i \in \mathcal{C}_t} u_{ix_t}(t)$
13:     Play $(x_t, y_t)$. Receive preference $o_t$
14:     Update: $\forall i, j \in [K]$, $w_{x_t y_t}(t+1) \leftarrow w_{x_t y_t}(t) + o_t$, $w_{y_t x_t}(t+1) \leftarrow w_{y_t x_t}(t) + (1 - o_t)$,
      $a_{ij}(t+1) \leftarrow \max\{n_{ij}(t+1), C(K, \delta)\}, \forall i, j \in [K]$
15: **end for**

---

## 4   `SlDB-UCB`**: A Fixed-Confidence Algorithm**

In this section, we design an efficient algorithm for the `DB-SPAA`$(K, T)$ problem with instance-dependent regret guarantee.

**Main ideas.** Our algorithm, described in Alg. 1, depends on an hyper-parameter $\alpha > 0.5$ and a confidence parameter $\delta > 0$. It maintains, for each item $k \in [K]$, its own record of empirical pairwise estimates of the duels, $(i, j) \in [K] \times [K]$ and their respective upper confidence bounds defined as:

$$\widehat{p}_{ij}(t) := \frac{w_{ij}(t)}{n_{ij}(t)} \qquad \text{and} \qquad u_{ij}(t) := \widehat{p}_{ij}(t) + c_{ij}(t), \ \text{with} \quad c_{ij}(t) := \sqrt{\frac{\alpha \log a_{ij}(t)}{n_{ij}(t)}},$$

where $w_{ij}(t)$ denotes the total number of times item $i$ beats $j$ up to round $t$, $n_{ij}(t) := w_{ij}(t) + w_{ji}(t)$, and for all $i, j \in [K]$ and $t \in [T]$

$$a_{ij}(t) := \max\{C(K, \delta), n_{ij}(t)\} \qquad \text{and} \qquad C(K, \delta) := \left(\frac{(4\alpha-1)K^2}{(2\alpha-1)\delta}\right)^{\frac{1}{2\alpha-1}}.$$

A key observation is that our careful choice of the confidence bounds $c_{ij}(t)$ ensures that with high probability $p_{ij}(t) \in [\widehat{p}_{ij}(t) - c_{ij}(t), \widehat{p}_{ij} + c_{ij}(t)]$ for any duel $i, j \in [K]$ and any $t \in [T]$ (Lem. 4). Now at any round $t \geq 1$, the algorithm first computes a set of potential winners of $S_t$ as $\mathcal{C}_t = \{k \in S_t \mid |\mathcal{C}_k(t)| = \max_{j \in S_t} |\mathcal{C}_j(t)|\}$, where $\mathcal{C}_k(t) := \{j \in S_t \mid u_{kj}(t) > \frac{1}{2}\}$ denotes the set of items that item $k$ dominates (optimistically). At each round, we play a random item from the set potential winners $\mathcal{C}_t$ as the left arm $x_t$. Finally the right-arm $y_t$ is chosen to be the most competitive opponent of $x_t$ as $y_t \leftarrow \arg\max_{i \in \mathcal{C}_t} u_{ji}(t)$ from the potential winners. Our arm selection strategy ensures that eventually for all $t$, algorithm plays the optimal pair $(i_t^*, i_t^*)$ frequently enough as desired.

**Theorem 3** (Fixed-confidence regret analysis: `SlDB-UCB`)**.** *Given any $\delta > 0$ and $\alpha > 1/2$, with probability at least $1 - \delta$, the regret incurred by $SlDB\text{-}UCB$ (Alg. 1) is upper-bounded as:*

$$R_T \leq 2 \sum_{i=1}^{K-1} \sum_{j=i+1}^{K} M_{ij} \log\left(2C(K, \delta) M_{ij}\right)$$

*where*

$$C(K, \delta) := \left(\frac{(4\alpha - 1)K^2}{(2\alpha - 1)\delta}\right)^{\frac{1}{2\alpha-1}} \quad \text{and} \quad M_{ij} = \sum_{k=1}^{i} \frac{4\alpha}{\min\left\{\Delta(k, i)_+, \Delta(k, j)_+\right\}^2}.$$

The complete proof with a precise dependencies on the model parameters is deferred to Appendix 9.

**Remark 2.** *The dependency on $\Delta = \min_{i,j} \Delta_+(i, j)$ does not match the lower-bound of Thm. 1, which is of order $O(\log(T)/\Delta)$. Instead, Thm. 3 proves $O(\log(1/\delta)/\Delta^2)$. Yet, the bounds are*

*not directly comparable because the lower-bound is on the expected regret while the upper-bound considers fixed-confidence $\delta$ and is hence independent of $T$. All existing dueling bandit algorithms, that minimize the expected regret, suffer an additional constant term of order $O(1/\Delta^2)$ –see for instance [37, 21]. Achieving an order $O(1/\Delta)$ dependendence is an interesting question for future work.*

*Proof sketch of Thm. 3.* The key steps lie in proving the following four lemmas. The first lemma follows along the line of Lem. 1 of RUCB algorithm [37]and shows that all the pairwise estimates are contained within their respective confidence intervals with high probability.

**Lemma 4.** *Let $\alpha > 0.5$ and $\delta > 0$. Then, for any $i, j \in [K]$, with probability at least $1 - \delta/K^2$,*

$$\widehat{p}_{ij}(t) - c_{ij}(t) \leq p_{ij} \leq u_{ij}(t) := \widehat{p}_{ij}(t) + c_{ij}(t), \qquad \forall t \in [T] \,.$$

The lemma below shows that once the algorithm can not play any suboptimal pair 'too many times'.

**Lemma 5.** *Let $\alpha > 0.5$. Under the notations and the high-probability event of Lem. 4, for all $i, j, k \in [K]$ such that $\{i, j\} \neq \{k, k\}$, and for any $\tau \geq 1$*

$$\sum_{t=1}^{\tau} \mathbf{1}(i_t^* = k)\mathbf{1}\big(\{x_t, y_t\} = \{i, j\}\big) \leq \frac{4\alpha \log a_{i,j}(\tau)}{\min\big\{\Delta(k,i)_+, \Delta(k,j)_+\big\}^2} \,,$$

*where recall $a_{ij}(\tau) = \max\big(C(K, \delta), n_{ij}(\tau)\big)$.*

With probability of at least $1 - \delta$, the event of Lem. 4 holds and thus so do the ones of Lem. 5. The regret of Alg. 1 then follows from applying the above lemmas with the following careful decomposition of the regret:

$$R_T \leq \sum_{i=1}^{K-1} \sum_{j=i+1}^{K} \sum_{t=1}^{T} \sum_{k=1}^{i} \mathbf{1}(i_t^* = k)\mathbf{1}\big(\{x_t, y_t\} = \{i, j\}\big) = \sum_{i=1}^{K-1} \sum_{j=i+1}^{K} n_{ij}(T)$$

and the proof is concluded by using Lemma 5 to upper-bound $n_{ij}(T)$. The complete proof given in Appendix 9. □

## 5 `S1DB-ED`: An Expected Regret Algorithm

In this section, we propose another computationally efficient algorithm, `S1DB-ED` (Alg. 2), which achieves near-optimal expected-regret for `DB-SPAA` problem, and also performs competitively against `S1DB-UCB` empirically (see Sec. 6). Furthermore, a novelty of our finite time regret analysis of `S1DB-ED` lies in showing a cleaner trade-off between regret vs availability sequence $\mathcal{S}_T$ which automatically adapts to the inherent 'hardness' of the sequence of available subsets $\mathcal{S}_T$, unlike the previous attempts made in standard sleeping bandits for adversarial availabilities [19] (Rem. 3).

**Main ideas.** We again use the notations $w_{ij}(t), n_{ij}(t)$ as used for `S1DB-UCB` (Alg. 1), with the same initializations. Same as `S1DB-UCB`, this algorithm also maintains the empirical pairwise preferences $\widehat{p}_{ij}(t)$ for each item pair $i, j \in [K]$. However, unlike the earlier case here we need to ensure an initial $t_0$ rounds of exploration ($t_0 = 1$ in the theorem) for every distinct pairs $(i, j)$, and instead of maintaining pairwise UCBs, in this case the set of 'good-items' is defined in terms of *empirical divergences* for all $i \in S_t$

$$\mathcal{I}_i(t) := \sum_{j \in \widehat{\mathcal{B}}_i(t)} n_{ij}(t) \operatorname{kl}\big(\widehat{p}_{ij}(t), \nicefrac{1}{2}\big), \quad \widehat{\mathcal{B}}_i(t) := \Big\{j \in [K] \setminus \{i\} \mid \widehat{p}_{ij}(t) \leq \nicefrac{1}{2}\Big\} \cap S_t$$

denotes the empirical winners of item $i$ in set $S_t$. Now intuitively since $\exp(-\mathcal{I}_i(t))$ can be interpreted as the likelihood of $i$ being the *best-item* of $S_t$, we denote by $\widehat{i}_t^* \leftarrow \arg\min_{i \in S_t} \mathcal{I}_i(t)$ the *empirical-best* item of round $t$ and define the set of 'near-best' items $\mathcal{C}_t := \big\{i \in S_t \mid \mathcal{I}_i(t) - \mathcal{I}_{\widehat{i}_t^*}(t) \leq \alpha \log t\big\}$, whose likelihood is close enough to that of $\widehat{i}_t^*$. Finally the algorithm selects an arm pair $(x_t, y_t)$ such that $x_t$ is a potential candidate of good arm (which ensures the required exploration) and $y_t$ being the strongest challenger of $x_t$ w.r.t the empirical preferences. The algorithm is given in Alg. 2.

---

**Algorithm 2** `SlDB-ED`

---

1: **input:** Arm set: $[K]$, exploration parameter $t_0 > 0$, parameter $\alpha > 0$
2: **for** $t = 1, 2, \ldots, T$ **do**
3:    $\widehat{p}_{ij}(t) := \frac{w_{ij}(t)}{n_{ij}(t)}$, $\widehat{p}(i,i) \leftarrow \nicefrac{1}{2}$, $\forall i, j \in [K]$ (assume $\frac{x}{0} := 0.5$, $\forall x \in \mathbb{R}$)
4:    Receive $S_t \subseteq [K]$
5:    **if** $|S_t| \geq 2$ and $\exists i, j \in S_t$ s.t. $n_{ij}(t) < t_0$, $i \neq j$ **then**
6:       Set $x_t \leftarrow i$, $y_t \leftarrow j$          ▷ Exploration rounds
7:    **else**
8:       $\widehat{\mathcal{B}}_i(t) := \{j \in [K] \setminus \{i\} \mid \widehat{p}_{ij}(t) \leq \nicefrac{1}{2}\} \cap S_t$, $\forall i \in [K]$   ▷ Empirical winners over $i$
9:       $\mathcal{I}_i(t) := \sum_{j \in \widehat{\mathcal{B}}_i(t)} n_{ij}(t) \, \mathrm{kl}(\widehat{p}_{ij}(t), \nicefrac{1}{2})$, $\forall i \in [K]$ and $\widehat{i}_t^* \leftarrow \arg\min_{i \in S_t} \mathcal{I}_i(t)$
10:       $\mathcal{C}_t := \{i \in S_t \mid \mathcal{I}_i(t) - \mathcal{I}_{\widehat{i}_t^*}(t) \leq \alpha \log t\}$       ▷ Potential good arms
11:       Select any $x_t$ from $\mathcal{C}_t$ uniformly at random
12:       **if** $\left( \widehat{i}_t^* \in \widehat{\mathcal{B}}_{x_t}(t) \text{ or } \widehat{\mathcal{B}}_{x_t}(t) = \emptyset \right)$: set $y_t \leftarrow \widehat{i}_t^*$, **else:** $y_t \leftarrow \arg\max_{i \in S_t \setminus \{x_t\}} \widehat{p}_{ix_t}(t)$
13:    **end if**
14:    Play $(x_t, y_t)$ Receive preference feedback $o_t$
15: **end for**

---

**Theorem 6** (Expected regret analysis `SlDB-ED`). *Let $t_0 = 1$ and $\alpha = 4K$. Then as $T \to \infty$, the expected regret incurred by SlDB-ED (Alg. 2) can be upper bounded as: For all $\varepsilon_2, \ldots, \varepsilon_K \geq 0$*

$$\mathbf{E}\big[R_T\big] \lesssim K^2 + \sum_{1 \leq i < j \leq K} \left( \frac{K \mathbf{1}_{\{\Delta(i,j) > \varepsilon_j\}}}{\Delta(i,j)^2} + n_{ij}(T) \min\{\varepsilon_j, \Delta(i,j)\} \right) + \sum_{j=2}^{K} \frac{K \log T}{\max\left\{\varepsilon_j, \Delta(j-1,j)_+\right\}}$$

$$\leq O\left( \min\left\{ \sum_{j=2}^{K} \frac{K \log T}{\Delta(j-1,j)_+}, \ K T^{2/3} \right\} \right).$$

The proof is deferred to Appendix 10. Although it borrows some high-level ideas from [19] for sleeping bandits and from [21] for RMED in standard dueling bandits, our analysis needed new ingredients in order to obtain $O(K^2(\log T)/\Delta)$. This is especially the case for the proofs of the technical Lemmas 8 and 9 which significantly differ from "corresponding" technical lemmas of [21]. Specifically, both regret bounds of RMED and ours need to control the length of an initial exploration $t_0$ after which pairwise preferences are well estimated by $\widehat{p}_{ij}(t)$. This is done respectively by our Lemma 8 and Lemma 5 of [21]. Yet, RMED's original analysis is based on a union bound over all possible subsets $S \subset \{1, \ldots, K\}$ of items (see Equation (19) in [21]), whose number is exponential in $K$. Despite our efforts, we could not follow the proof of Lemma 5 of [21], which to the best of our understanding, should yield to an exploration $t_0$ exponentially large in $K$ contrary to $O(1)$ claimed in [21]. Instead, in our proof of Lem. 8, we carefully apply concentration inequalities to run union bounds over the items directly instead of sets of items.

The upper-bound of Thm. 6 is close to optimal. It suffers at most a suboptimal factor $K$ and exactly matches the lower-bound for some problems. The distribution-free upper-bound of order $O(T^{2/3})$ matches obtainable standard dueling bandit problems [21, 37], since the latter algorithms also suffer constant terms of order $\Delta^{-2}$. Yet, it is unclear whether it is optimal or if $O(\sqrt{T})$ can be obtained.

**Remark 3** (Sequence $\mathcal{S}_T$ adaptivity of Alg. 2)**.** It is worth pointing out that the regret bound of Thm. 6 is finite time and automatically adapts to the sequence of available sets $\mathcal{S}_T$. In the worst-case, the complexity lies in identifying for all items $j$ the gap with the earlier item $j-1$. Yet, our regret-bound, which holds for any $\varepsilon_j \geq 0$, will automatically perform a trade-off for each $j$ between the gap $\Delta(j-1,j)_+^{-1}$ and $\varepsilon_j \sum_{i=1}^{j-1} n_{ij}(T)$ the number of times $j$ is played together with a better item $i < j$. In particular, $\sum_{i=1}^{j-1} n_{ij}(T)$ can be small if $j$ is rarely available in $S_t$ while not optimal. Notably, this adaptivity to $\mathcal{S}_T$ item per item improves the regret guarantee of Thm. (10), [19], which also addresses the problem of sleeping bandits with 'adversarial availabilities' but for the stochastic multi-armed bandit setup and only provides worst-case guarantees over all $\mathcal{S}_T$ and a trade-off $\varepsilon$ independent of $j$.

**Remark 4** (`SlDB-ED` in standard dueling bandits)**.** Even in the dueling bandit setting (without the sleeping component), `SlDB-ED` and Thm. 6 have advantages compared to the RMED algorithm and analysis of [21]. Our regret bound is valid for all number of items $K$, while the one of Thm. 3 of [21] is only asymptotic when $K \to \infty$. This is due to the fact that the algorithm of [21] depends

on a hyper-parameter $f(K)$ which needs to larger than $AK$, where $A$ is a constant in $K$ and $T$ but which depends on the unknown sub-optimality gaps $\Delta(i,j)$. Thus, [21] chooses $f(K) \approx K^{1+\varepsilon}$ so that eventually the bound is satisfied when $K \to \infty$. Instead, our algorithm only depends on easily tunable hyper-parameters $t_0$ and $\alpha$, whose optimal values are independent of unknown parameters.

## 6 Experiments

In this section, we compare the empirical performances of our two proposed algorithms (Alg. 1 and 2). Note that there are no other existing algorithms for our problem (see Sec. 1).

**Constructing Preference Matrices (P).** We use the following three different utility based Plackett-Luce($\boldsymbol{\theta}$) preference models (see Sec. 2) that ensures a *total-ordering*. We now construct three types of problem instances 1. *Easy* 2. *Medium* 3. *Hard*, for any given $K$, such that items with their respective $\boldsymbol{\theta}$ parameters are assigned as follows: 1. *Easy:* $\boldsymbol{\theta}(1 : \lfloor K/2 \rfloor) = 1, \boldsymbol{\theta}(\lfloor K/2 \rfloor + 1 : K) = 0.5$. 2. *Medium:* $\boldsymbol{\theta}(1 : \lfloor K/3 \rfloor) = 1, \boldsymbol{\theta}(\lfloor K/3 \rfloor + 1 : \lfloor 2K/3 \rfloor) = 0.7, \boldsymbol{\theta}(\lfloor 2K/3 \rfloor + 1 : K) = 0.4$. 3. *Hard:* $\boldsymbol{\theta}(i) = 1 - (i-1)/K, \forall i \in [K]$. Note for each $\boldsymbol{\sigma}^* = (1 > 2 > \ldots K)$.

In every experiment, we set the learning parameters $\alpha = 0.51, \delta = 1/T$ for `SlDB-UCB` (Alg. 1) and as per Thm. 6 for `SlDB-ED` (Alg. 2). All results are averaged over 50 runs.

**Regret over Varying Preference Matrices.** We first plot the cumulative regret of our two algorithms (Alg. 1 and 2) over time on the above three Plackett-Luce datasets for K = 10. We generate availability sequence $\mathcal{S}_T$ randomly by sampling every item $i \in [K]$ independently with probability $0.5$. Fig 2 shows that, as their names suggest too, *instance-Easy* is easiest to learn as the best-vs-worst item preferences are well separated and the diversity of the item preferences across different groups are least. Consequently the algorithms yield slightly more regret on *instance-Medium* due to higher preference diversity, and the hardest instance being *Hard* where the learner really needs to differentiate the ranking of every item for any arbitrary set sequences $\mathcal{S}_T$. Empirically `SlDB-UCB` is seen to slightly outperform `SlDB-ED`, though orderwise they perform competitively.

**Regret over Varying Set Availabilities.** In these set of experiments, the idea is to understand how the regret improves over completely random subset availabilities as now the learner may not have to distinguish all item preferences as some of the item combinations occurs rarely. We choose $K = 10$ and to enforce item dependencies we generate each set $S_t$ by drawing a random sample from $\text{Gaussian}(\boldsymbol{\mu}, \Sigma)$ such that $\boldsymbol{\mu}_i = 0, \forall i \in [10]$, and $\Sigma$ is a fixed $10 \times 10$ positive definite matrix which controls the set dependencies: Precisely we use two different block diagonal matrices for *Low-Correlation* and *High-Correlation* with the following correlations: 1. *Low-Correlation:* $\Sigma$ is a separated block diagonal matrix on item partitions $\{1, 2, 3\}, \{4, 5, 6\}, \{7, 8, 9, 10\}$. 2. *High-Correlation:* $\Sigma$ is constructed by merging three all-1 matrices on partitions $\{1 \ldots 5\}, \{2, \ldots 8\}$, and $\{6, \ldots 10\}$, however as the resulting matrix is positive semi-definite, so we further take its SVD and reconstruct the matrix back eliminating the negative eigenvalues. At every round we sample a random vector from $\text{Gaussian}(\boldsymbol{\mu}, \Sigma)$, and $S_t$ is considered to be the set of items whose value exceeds $0.5$. Both experiments are run on *instance-Hard*. Fig. 3 shows, as expected, on *Low-Correlation* both algorithms converge to $\boldsymbol{\sigma}^*$ relatively faster and at lower regret compared to *High-Correlation* (as the latter induces higher variability of the available subsets).

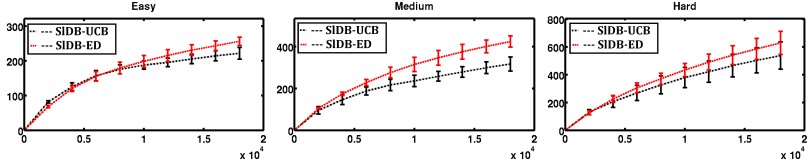

Figure 2: Regret ($R_t$) vs time ($t$) over three preference instances (**P**)

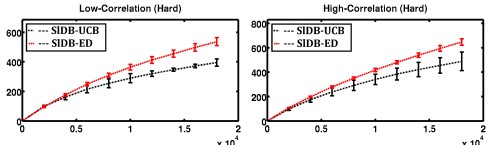
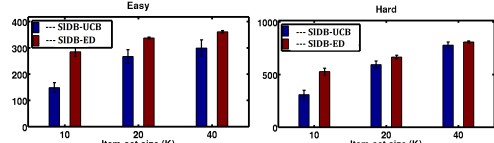

Figure 3: Regret ($R_t$) vs time ($t$) over availability sequences $\mathcal{S}_T$

Figure 4: Final regret ($R_T$) at $T = 10^4$ with varying sizes ($K$)

**Final Regret vs Setsize(K).** We also compared the (averaged) final regret of the two algorithms over varying item sizes $K$. We additionally constructed two larger Plackett-Luce ($\boldsymbol{\theta}$) *Easy* and *Hard* instances for $K = 20$ and $40$, using similar $\boldsymbol{\theta}$ assignments explained before. We set $T = 10\,000$ and use itemwise independent set generation idea, as described for Fig. 2. As expected, Fig. 4 shows the regret of both algorithms scales up with increasing $K$ with effect on `SlDB-ED` being slightly worse than `SlDB-UCB`, though the latter generally exhibits a higher variance.

**Worst Case Regret vs Time.** We run an experiment to analyse the regret of our two algorithms on the worst case problem instances. Towards this we use preference matrices $\mathbf{P}_\Delta$ of the form: $\mathbf{P}_\Delta(i,j) = 0.5 + \Delta, \forall 1 \le i < j \le K$, i.e. all items are spaced with equidistant gap $\Delta \in (0, 0.5]$. As before, we choose $T = 20,000$ and $K = 10$, and run the algorithms on above problem instances varying $\Delta$ in the range $[10/T, \dots, 0.5]$ with uniform grid-size of $0.005$ (i.e. total $100$ values of $\Delta$, each corresponds to a separate problem instance $\mathbf{P}_\Delta$ with different 'gap-complexity'). At the end we plot the worst case regret of both the algorithms over time, by plotting $\max_\Delta R_t(P_\Delta)$ vs $t$. We run the experiments over three availability sequences: 1. Independent (as used in Fig. 2), 2. Low-Correlation, and 3. High-Correlation (as used in Fig. 3). As a consequence the resulting plots reflect the worst case (w.r.t. $\Delta$) performances of the algorithms, which seem to be scaling as $O(T^{2/3})$ for `SlDB-ED`, as conjectured to be its distribution free upper bound (see discussion after Thm. 6), and with a slightly lower rate for `SlDB-UCB`. Fig. 5 shows the comparative performances.

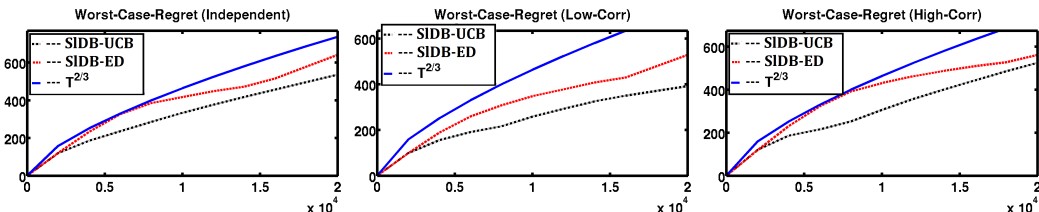

Figure 5: "Worst Case Regret" ($\max_\Delta R_t(P_\Delta)$) vs time ($t$) over three availability sequences $\mathcal{S}_T$

## 7 Conclusion and Perspective

We introduce the problem of sleeping dueling bandits with stochastic preferences and adversarial availabilities, which, despite of great practical relevance, was left unaddressed till date. Towards this we adapt two dueling bandit algorithms for the problem and give regret analysis for both. We also derive an instance dependent regret lower bound for our problem setup which shows that our second algorithm is asymptotically near-optimal (up to the problem dependent constants). Finally, we compare both our algorithms empirically where usually the first algorithm is shown to outperform the second, although having a relatively weaker regret.

**Future Works.** Moving forward, one can address many open questions along this direction, including relaxing the *total-ordering* assumption on the stochastic preferences assuming more general ranking objective based on borda [32] or copeland scores [36], or extending the framework to a general contextual scenario with subsetwise feedback. Another direction worth understanding is to analyze the connection of this problem with other bandit setups, e.g., learning with feedback graphs [3, 4] or other side information [23, 20]. It would also be interesting to consider the dueling bandit problem for adversarial preference and stochastic availabilities [24, 17], and also analyzing these class of problems for general subsetwise preferences [25, 30, 8].

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
