# Supplementary: Dueling Bandits with Adversarial Sleeping

## 8 Appendix for Sec. 3

**Definition 7** (*No-regret* algorithm). *An algorithm $\mathcal{A}$ for Sleeping-Dueling Bandit with Stochastic Preferences and Adversarial Availabilities problem is defined to be a No-regret algorithm, if for each problem instance* `DB-SPAA`$(\mathbf{P}, \mathcal{S}_T)$ *model, the expected number of times $\mathcal{A}$ plays any suboptimal duel $(i, j) \in [K] \times [K]$ is sublinear in T, or more precisely, $\forall (i,j) \neq (i_t^*, i_t^*)$, $\mathbf{E}[n_{ij}(T)] = o(T^\alpha)$, for some $\alpha \in (0, 1)$, where recall that we define $n_{ij}(t) := \sum_{\tau=1}^{t} \mathbf{1}\big(\{x_t, y_t\} = \{i, j\}\big)$ denotes the number of times the pair $(i, j)$ is played by $\mathcal{A}$ in T rounds. ($\mathbf{E}[\cdot]$ denotes expectation under the randomization of $\mathcal{A}$ and the* `DB-SPAA`$(\mathbf{P}, \mathcal{S}_T)$ *model.)*

### 8.1 Proof of Thm. 1

*Proof.* The main argument lies behind the fact that in the worst case the adversary can force the algorithm to learn the preference of every distinct pair $(i, j)$ as the in the 'worst-case' sequence $\mathcal{S}_T$ knowledge of the already 'learnt' pairwise preferences would not disclose any information on the remaining pairs; e.g. assuming $\boldsymbol{\sigma}^* = (1, 2, \ldots, K)$, revealing the available subsets in the following sequence $(1, 2), (1, 3), \ldots (1, K), (2, 3), (2, 4), \ldots (K-1, K)$ would force the learner to explore (learn the preferences) all $\binom{K}{2}$ distinct pairs.

The remaining proof establishes this formally, towards which we first show a $\Omega(\frac{\ln T}{\Delta(1,2)})$ regret lower bound for a `DB-SPAA` instance with just two items (i.e. $K = 2$) as shown in Lem. 2. The lower bound for any general $K$ can now be derived applying the above bound on independent $\binom{K}{2}$ subintervals, with the availability sequence $(1, 2), (1, 3), \ldots (1, K), (2, 3), (2, 4), \ldots (K-1, K)$.

For the interest of the problem instance construction to prove the lower bound, we would assume $\Delta(i, i+1) > 0$, $\forall i \in [K-1]$ and thus we use $\Delta(i, i+1)_+ = \Delta(i, i+1)$ for the rest of this proof (as also assumed for Lem. 2). Note that this is without loss of generality since otherwise the regret lower bound in Lem. 2 is trivially $0$.

We add the details below for completeness.

Let $K' = \binom{K}{2}$ and suppose we divide the time horizon into sub-intervals $1, 2, \ldots K'$ each of length $T' := T/K'$, where the available subsets are fixed inside every subinterval, and follows the sequence $(1, 2), (1, 3), \ldots (1, K), (2, 3), (2, 4), \ldots (K-1, K)$ across subintervals. Note that with above construction, the regret minimization problem within each sub-interval boils down to the standard stochastic dueling bandit problem over 2 arms.

Further since the preferences of available set $S_t$s are independent across different sub-intervals, applying the lower bound of Lem. 2 individually to every $K'$ subintervals the total cumulative regret of $\mathcal{A}$ in $T$ rounds can be lower bounded as:

$$\mathbf{E}[R_T(\mathcal{A})] = \sum_{i=1}^{K-1} \sum_{j=i+1}^{K} \mathbf{E}[R_{T'}(\mathcal{A})] \geq \Omega\bigg(\sum_{i=1}^{K-1} \sum_{j=i+1}^{K} \frac{\log T'}{\Delta(i,j)}\bigg) = \Omega\bigg(\sum_{i=1}^{K-1} \sum_{j=i+1}^{K} \frac{\log T}{\Delta(i,j)}\bigg)$$

where the last inequality holds since $T \geq (K')^2$, which implies $\log \frac{T}{K'} \geq \log T - \log \sqrt{T} = \frac{1}{2} \log T$, and this concludes the proof.

$\square$

**Lemma 2** (Lower Bound of `DB-SPAA`$(\mathbf{P}_K, \mathcal{S}_T)$ for 2 items). *For any No-regret learning algorithm $\mathcal{A}$, there exists a problem instance* `DB-SPAA`$(\mathbf{P}_2, \mathcal{S}_T)$ *such that the expected regret incurred by $\mathcal{A}$ on that can be lower bounded as: $\mathbf{E}[R_T(\mathcal{A})] \geq \Delta^{-1} \log(T)$, $\Delta$ being the 'preference-gap' between the two items (i.e. $\Delta = \mathbf{P}_{12} - \frac{1}{2}$, assuming $P_{12} > \frac{1}{2}$ or equivalently $\Delta > 0$).*

*Proof.* Note that for $K = 2$, the only non-trivial available set is $\{1, 2\}$, therefore we assume $S_t = \{1, 2\}$, $\forall t \in [T]$. The proof now simply follows by applying the existing lower bound (Thm. 2) of [34] for standard stochastic dueling bandit problem for only 2 arms. $\qquad\square$

# 9    Appendix for Sec. 4

**Notations.** Let us start with defining useful notation for the analysis. We write for any pair $1 \le i < j \le K$

$$M_{ij} = \sum_{k=1}^{i} \frac{4\alpha}{\min\left\{\Delta(k,i)_+, \Delta(k,j)_+\right\}^2}.$$

We also denote $S_{\backslash i} = S \setminus \{i\}$, $i \in S$, for any $S \subseteq [K]$.

## 9.1   Complete proof of Thm. 3

**Theorem 3.** *Given any $\delta > 0$ and $\alpha \ge 1$, with probability at least $1 - \delta$, the regret incurred by* `SlDB-UCB` *(Alg. 1) is upper-bounded as:*

$$R_T \le 2 \sum_{i=1}^{K-1} \sum_{j=i+1}^{K} M_{ij} \log\left(2C(K,\delta)M_{ij}\right)$$

*where* $C(K,\delta) := \left((4\alpha - 1)K^2/((2\alpha - 1)\delta)\right)^{\frac{1}{2\alpha - 1}}$.

*Proof of Thm. 3.* The key steps lie in proving the following four lemmas. The first lemma follows along the line of Lem. 1 of RUCB algorithm [37]. It shows after $C(K,\delta)$ rounds all the pairwise estimates are contained within their respective confidence intervals:

**Lemma 4.** *Let $\alpha > 0.5$ and $\delta > 0$. Then, with probability at least $1 - \delta$, for any $i, j \in [K]$*

$$\widehat{p}_{ij}(t) - c_{ij}(t) \le p_{ij} \le u_{ij}(t) := \widehat{p}_{ij}(t) + c_{ij}(t), \qquad \forall t \in [T].$$

The lemma below is adapted from Proposition 2, [37]. It basically states that once the algorithm has explored enough (i.e., more than $C(K,\delta)$) the algorithm will not play a suboptimal pair too many times.

**Lemma 5.** *Let $\alpha > 0.5$. Under the notations and the high-probability event of Lem. 4, for all $i, j, k \in [K]$ such that $\{i, j\} \neq \{k, k\}$, and for any $\tau \ge 1$*

$$\sum_{t=1}^{\tau} \mathbf{1}(i_t^* = k)\mathbf{1}\left(\{x_t, y_t\} = \{i, j\}\right) \le \frac{4\alpha \log a_{i,j}(\tau)}{\min\left\{\Delta(k,i)_+, \Delta(k,j)_+\right\}^2},$$

*where recall $a_{ij}(\tau) = \max\left(C(K,\delta), n_{ij}(\tau)\right)$.*

Given the above results, we are ready to analyze the regret guarantee of `SlDB-UCB`. For ease on notation we denote $\mathcal{X}_t = \{x_t, y_t\}$. Let us assume the 'good event' of Lem. 4 holds good for all $t \in [T]$, which is true with probability of at least $1 - \delta$. Conditioned on that, note that Lem. 5 is satisfied. Based on this we now analyze the regret of Alg. 1:

$$
\begin{aligned}
R_T &= \sum_{t=1}^{T} \sum_{k=1}^{K-1} \mathbf{1}(i_t^* = k) r_t \\
&= \sum_{t=1}^{T} \sum_{k=1}^{K} \sum_{i=k}^{K} \sum_{j=i}^{K} \mathbf{1}(i_t^* = k)\mathbf{1}\left(\{x_t, y_t\} = \{i, j\}\right) r_t \qquad \leftarrow \text{ because } x_t \ge k \text{ and } y_t \ge k \\
&= \sum_{t=1}^{T} \sum_{k=1}^{K-1} \sum_{i=k}^{K-1} \sum_{j=i+1}^{K} \mathbf{1}(i_t^* = k)\mathbf{1}\left(\{x_t, y_t\} = \{i, j\}\right) r_t \leftarrow \text{ because } i = j = k \text{ implies } r_t = 0
\end{aligned}
$$

$$\leq \sum_{t=1}^{T} \sum_{k=1}^{K-1} \sum_{i=k}^{K-1} \sum_{j=i+1}^{K} \mathbf{1}(i_t^* = k)\mathbf{1}\big(\{x_t, y_t\} = \{i,j\}\big) \quad \leftarrow \text{ because } r_t \leq 1$$

$$= \sum_{i=1}^{K-1} \sum_{j=i+1}^{K} \sum_{t=1}^{T} \sum_{k=1}^{i} \mathbf{1}(i_t^* = k)\mathbf{1}\big(\{x_t, y_t\} = \{i,j\}\big)$$

$$= \sum_{i=1}^{K-1} \sum_{j=i+1}^{K} n_{ij}(T). \tag{3}$$

Now, fix $1 \leq i < j \leq K$ and let us upper-bound $n_{ij}(T)$ the number of times such a pair is played. Summing the upper-bound of Lemma 5 over $k \leq i$, we get

$$n_{ij}(T) = \sum_{k=1}^{i} \sum_{t=1}^{T} \mathbf{1}(i_t^* = k)\mathbf{1}\big(\{x_t, y_t\} = \{i,j\}\big) \leq \sum_{k=1}^{i} \frac{4\alpha \log(\max\{C(K,\delta), n_{ij}(T)\})}{\min\left\{\Delta(k,i)_+, \Delta(k,j)_+\right\}^2}.$$

Therefore, since $C(K,\delta) \geq 1$,

$$n_{ij}(T) \leq M_{ij}\big(\log(C(K,\delta) + \log(n_{ij}(T)))\big), \qquad \text{where} \quad M_{ij} = \sum_{k=1}^{i} \frac{4\alpha}{\min\left\{\Delta(k,i)_+, \Delta(k,j)_+\right\}^2}.$$

which implies

$$n_{ij}(T) \leq 2M_{ij}\big(\log C(K,\delta) + \log(2M_{ij})\big).$$

Substituting into Inequality (3) entails

$$R_T \leq 2 \sum_{i=1}^{K-1} \sum_{j=i+1}^{K} M_{ij} \log\big(2C(K,\delta)M_{ij}\big),$$

which concludes the proof. $\qquad\qquad\qquad\qquad\qquad\qquad\qquad\qquad\qquad\qquad\qquad\qquad\qquad\square$

## 9.2 Technical lemmas for Thm. 3

**Lemma 4.** *Let $\alpha > 0.5$ and $\delta > 0$. Then, with probability at least $1 - \delta$, for any $i, j \in [K]$*

$$\widehat{p}_{ij}(t) - c_{ij}(t) \leq p_{ij} \leq u_{ij}(t) := \widehat{p}_{ij}(t) + c_{ij}(t), \qquad \forall t \in [T].$$

*Proof.* The proof of this lemma is adapted from a similar result (Lemma 1) of [37]. Suppose $\mathcal{G}_{ij}(t)$ denotes the event that at time $t \in [T]$ and item-pair $i, j \in [K]$, $p_{ij} \in [l_{ij}(t), u_{ij}(t)]$. We also define $\mathcal{G}_{ij}^c(t)$ its complement. Let $i, j \in [K]$.

Note that for any such that pair $(i,i)$, $\mathcal{G}_{ii}(t)$ always holds true for any $t \in [T]$ and $i \in [n]$, as $p_{ii} = u_{ii} = l_{ii} = \frac{1}{2}$. We can thus assume $i \neq j$. Moreover, for any $t$ and $i, j$, $\mathcal{G}_{ij}(t)$ holds if and only if $\mathcal{G}_{ij}(t)$ as $|\widehat{p}_{ji}(t) - p_{ji}| = |(1 - \widehat{p}_{ij}(t)) - (1 - p_{ij})| = |\widehat{p}_{ij}(t) - p_{ij}|$. Thus we will restrict our focus only to pairs $i < j$ for the rest of the proof. Hence, to prove the lemma it suffices to show

$$\mathbf{P}\Big(\exists t \in [T], i < j, \text{ such that } \mathcal{G}_{ij}^c(t)\Big) \leq \delta,$$

which we do now. Recall from the definition of $c_{ij}(t)$ that $\mathcal{G}_{ij}(t)$ can be rewritten as:

$$|\widehat{p}_{ij}(t) - p_{ij}| \leq \sqrt{\frac{\alpha \ln(a_{ij}(t))}{n_{ij}(t)}}.$$

Let $\tau_{ij}(n)$ the time step $t \in [T]$ when the pair $(i,j)$ was updated (i.e. $i$ and $j$ was compared) for the $n^{th}$ time. We now bound the probability of the confidence bound ($\mathcal{G}_{ij}(t)$) getting violated at any round $t \in [T]$ for some duel $(i,j)$ as follows:

$$\mathbf{P}\Big(\exists t \in [T], i < j, \text{ such that } \mathcal{G}_{ij}^c(t)\Big) \leq \sum_{i<j} \mathbf{P}\left(\exists n \geq 0, |p_{ij} - \widehat{p}_{ij}(\tau_{ij}(n))| > \sqrt{\frac{\alpha a_{ij}(\tau_{ij}(n))}{n_{ij}(\tau_{ij}(n))}}\right)$$

$$= \sum_{i<j} \left[ \mathbf{P}\left( \exists n \le C(K,\delta), \ |p_{ij} - \widehat{p}_{ij}(n)| > \sqrt{\frac{\alpha \ln(C(K,\delta))}{n}} \right) \right.$$

$$\left. + \mathbf{P}\left( \exists n > C(K,\delta), \ |p_{ij} - \widehat{p}_{ij}(\tau_{ij}(n))| > \sqrt{\frac{\alpha \ln\left(n_{ij}\left(\tau_{ij}(n)\right)\right)}{n}} \right) \right],$$

where $\widehat{p}_{ij}(t) = \frac{w_{ij}(t)}{w_{ij}(t) + w_{ij}(t)}$ is the frequentist estimate of $p_{ij}$ at round $t$ (after $n = n_{ij}(t)$ comparisons between arm $i$ and $j$). To ease the notation, denote $F = C(K,\delta)$. Noting $n_{ij}(\tau_{ij}(n)) = n$, and using Hoeffding's inequality, we further get

$$\mathbf{P}\left( \exists t \in [T], i < j, \text{ such that } \mathcal{G}_{ij}^c(t) \right) \le \sum_{i<j} \left[ \sum_{n=1}^{F} 2e^{-2n\frac{\alpha \ln F}{n}} + \sum_{n=F+1}^{\infty} 2e^{-2n\frac{\alpha \ln n}{n}} \right]$$

$$= \frac{n(n-1)}{2} \left[ 2\sum_{n=1}^{F} \frac{1}{F^{2\alpha}} + \sum_{n=F+1}^{\infty} \frac{2}{n^{2\alpha}} \right]$$

$$\le \frac{n^2}{F^{2\alpha-1}} + n^2 \int_F^{\infty} \frac{dx}{x^{2\alpha}} \le \frac{n^2}{F^{2\alpha-1}} - \frac{n^2}{(1-2\alpha)F^{2\alpha-1}} = \frac{(2\alpha)n^2}{(2\alpha-1)F^{2\alpha-1}} = \delta.$$

where the last inequality is because $F = C(K,\delta) = \left[ \frac{2\alpha n^2}{(2\alpha-1)\delta} \right]^{\frac{1}{2\alpha-1}}$. This concludes the claim. $\qquad\square$

**Lemma 5.** *Let $\alpha > 0.5$. Under the notations and the high-probability event of Lem. 4, for all $i, j, k \in [K]$ such that $\{i,j\} \ne \{k,k\}$, and for any $\tau \ge 1$*

$$\sum_{t=1}^{\tau} \mathbf{1}(i_t^* = k)\mathbf{1}\left(\{x_t, y_t\} = \{i,j\}\right) \le \frac{4\alpha \log a_{i,j}(\tau)}{\min\left\{\Delta(k,i)_+, \Delta(k,j)_+\right\}^2},$$

*where $a_{ij}(\tau) = \max\left(C(K,\delta), n_{ij}(\tau)\right)$.*

*Proof.* We assume the confidence bound of Lem. 4 is holds good for all pair $(i,j) \in [K]^2$, at all round $t \in [T]$, which we know happens with probability at least $(1-\delta)$. Let us define $l_{ij}(t) := 1 - u_{ji}(t)$. Let $t \ge 1$. Let $i, j, k \in [K]$ such that $i_t^* = k$, $x_t = i$, and $y_t = j$ and $\{i,j\} \ne \{k,k\}$. Since $i_t^* = k$, this implies both $i \ge k$ and $j \ge k$. Furthermore, we recall that $i_t^* = k$ is unique by definition, $i_t^* = \min\{S_t\}$. We consider the following cases.

- **Case 1 ($i = j > k$).** Then, $x_t = y_t = i = j$. By the arm selection strategy (Step 14. of Algorithm 1)

$$y_t \leftarrow \arg\max_{m \in \mathcal{C}_t} u_{mx_t}(t)$$

  which implies $1/2 = u_{jj}(t) > u_{kj}(t)$. But, on the other hand, since $k < j$, by Lemma 4, $u_{kj}(t) \ge p_{kj} > \frac{1}{2}$. This causes a contraction and this case is not possible.

- **Case 2 ($j > i = k$).** Then, $y_t = j$ and $x_t = i_t^* = k$. We again proceed by contradiction. Assume that $n_{kj}(t) > \frac{4\alpha \ln a_{kj}(t)}{\Delta(k,j)_+^2}$. Then, by definition of $c_{kj}(t)$, it implies

$$2c_{kj}(t) = 2\sqrt{\frac{\alpha \log a_{kj}(t)}{n_{ij}(t)}} < \Delta(k,j)_+,$$

  which by Lem. 4 entails

$$u_{jk}(t) = \widehat{p}_{jk}(t) + c_{jk}(t) < p_{jk} + 2c_{jk}(t) < \frac{1}{2} - \Delta(k,j)_+ + \Delta(k,j)_+ < \frac{1}{2}.$$

  Again since our arm selection strategy enforces $y_t \leftarrow \arg\max_{i \in \mathcal{C}_t} u_{ix_t}(t)$, clearly $\frac{1}{2} = u_{kk}(t) > u_{jk}(t)$, so that $j$ can not be selected as $y_t$. Therefore, recalling that $k = i$,

$$n_{ij}(t) \le \frac{4\alpha \ln a_{kj}(t)}{\Delta(i,j)_+^2}. \tag{4}$$

- **Case 3** ($i > j = k$). Then, $x_t = i$ and $y_t = i_t^* = k$. This can be proved similarly as the previous case. Assuming $n_{ik}(t) > 4\alpha \ln a_{ik}(t)\Delta(i,k)_+^{-2}$ yields $u_{ik}(t) < 1/2$. Therefore, since $u_{ii}(t) = 1/2$, it entails

$$|C_i(t)| = \left|\{m \in S_t | u_{im}(t) > 1/2\}\right| \le |S_t \backslash \{i,k\}| \le |S_t| - 2.$$

But by Lemma 4, for all $m > k$, $u_{km}(t) \ge p_{km} = 1/2 + \Delta(k,m) > 1/2$. Thus, since $k = i_t^*$, we also have $|C_k(t)| = |S_t| - 1$ and thus

$$|C_k(t)| > |C_i(t)|.$$

By Step 12 of Algorithm 1, this implies that $i \notin C_t$ and thus $x_t \ne i$ as $x_t$ is selected from $C_t$, which causes a contradiction. Therefore, recalling $j = k$,

$$n_{ij}(t) \le \frac{4\alpha \ln a_{i,j}(t)}{\Delta(i,j)_+^2}. \tag{5}$$

- **Case 4.** ($i \ne j > k$). Then, assuming $n_{ij}(t) > 4\alpha \log a_{i,j}(t) \min\{\Delta(k,i)_+, \Delta(k,j)_+\}^{-2}$, note that

$$u_{ij}(t) - l_{ij}(t) = 2c_{ij}(t) = 2\sqrt{\frac{\alpha \log a_{i,j}(t)}{n_{ij}(t)}} < \min(\Delta(k,i)_+, \Delta(k,j)_+).$$

But, on the other hand, $x_t = i$ implies $u_{ij}(t) > 1/2$, and $y_t = j$ implies $u_{ji}(t) > u_{jk}(t) > p_{jk}$, and then $l_{ij}(t) = 1 - u_{ji}(t) < 1 - p_{jk}$. So we have $u_{ij}(t) - l_{ij}(t) > 1/2 - p_{ik} = \Delta(k,i)_+$ which gives a contradiction. Thus,

$$n_{i,j}(t) \le \frac{4\alpha \log a_{i,j}(t)}{\min\left\{\Delta(k,i)_+, \Delta(k,j)_+\right\}^2}. \tag{6}$$

Note that the case $x_t = j$, $y_t = i$, and $i_t^* = k$ is symmetric with the above cases and can be considered similarly. Denote by $\tau'$ the last time before $\tau \ge 1$ such that a pair $\{i,j\} \ne \{k,k\}$ is pulled when $k = i_t^*$, that is

$$\tau' = \operatorname*{argmax}_{1 \le t \le \tau} \left\{k = i_t^*, \{x_t, y_t\} = \{i,j\}\right\}.$$

Then,

$$\sum_{t=1}^{\tau} \mathbf{1}(i_t^* = k)\mathbf{1}\big(\{x_t, y_t\} = \{i,j\}\big) \le \sum_{t=1}^{\tau'} \mathbf{1}(i_t^* = k)\mathbf{1}\big(\{x_t, y_t\} = \{i,j\}\big) \le n_{ij}(\tau').$$

But, at time $\tau'$, the suboptimal pair $\{i,j\}$ got pulled, thus one of the above four cases is true, which implies from Inequalities (4), (5), and (6) that

$$n_{ij}(\tau') \le \frac{4\alpha \log a_{i,j}(\tau)}{\min\left\{\Delta(k,i)_+, \Delta(k,j)_+\right\}^2}.$$

Substituting into the previous inequality concludes the proof. □

# 10  Appendix for Sec. 5

## 10.1  Technical lemmas

Before proving the regret guarantee of `S1DB-ED` (Alg. 2) in Thm. 6, we would like to introduce three lemmas which are crucially used towards bounding Alg. 2's regret. Lemma 8 below states that after some exploration, the algorithm estimates well all $p_{ij}$ with $\widehat{p}_{ij}(t)$.

**Lemma 8.** *Let $t_0 \ge 1$ and $\alpha \ge 4K$. Let $\varepsilon_i > 0$ for all $i \in [K]$. For any $t \in [T]$, let us define $\mathcal{E}_t := \{n_{ij}(t) > t_0, \forall i,j \in S_t, i \ne j\}$ to be the event when all distinct pairs $i,j \in [K]$ is played for at least for $t_0$ times. Let us also denote the event $\mathcal{G}(t) := \{\forall j > i_t^*, \Delta(i_t^*, j) > \varepsilon_i, \widehat{p}_{i_t^* j}(t) > 1/2\}$. $\mathcal{G}^c(t)$ denotes the complement event of $\mathcal{G}(t)$. Then `S1DB-ED` satisfies:*

$$\mathbf{E}\left[\sum_{t=1}^{T} \mathbf{1}(\mathcal{E}_t)\mathbf{1}\big(\mathcal{G}^c(t)\big)\right] = 2K + K \sum_{i=1}^{K-1} \sum_{j=i+1 | \Delta(i,j) > \varepsilon_i}^{K} \frac{e^{-(t_0-1)\Delta(i,j)^2}}{\Delta(i,j)^2}.$$

*Proof.* First, we show that with high probability for all $t = 1, \ldots, T$, $i_t^* \in \mathcal{C}_t$, $i_t^*$ belongs to the set of potential winners. Let $t \geq 1$. By definition of $\mathcal{C}_t$, we have

$$
\mathbf{P}(i_t^* \notin \mathcal{C}_t) \leq \mathbf{P}\Big( \sum_{j \in \widehat{\mathcal{B}}_{i_t^*}(t)} n_{i_t^* j}(t) \, \mathrm{kl}(\widehat{p}_{i_t^* j}(t), 0.5) \geq \alpha \log t \Big)
$$

$$
\leq \mathbf{P}\Big( \exists j > i_t^* \quad \text{s.t.} \quad n_{i_t^* j}(t) \, \mathrm{kl}(\widehat{p}_{i_t^* j}(t), 0.5) \geq \frac{\alpha \log t}{K} \Big)
$$

$$
\leq \sum_{j = i_t^* + 1}^{K} \sum_{n=1}^{t} \mathbf{P}\Big( \mathrm{kl}(\tilde{p}_{i_t^* j}(n), 0.5) \geq \frac{\alpha \log t}{n K} \Big),
$$

where $\tilde{p}_{ij}(n)$ denotes the frequentist empirical estimate of $P(i, j)$ after $n$ pairwise comparisons between $i$ and $j$ (i.e., $\tilde{p}_{ij}(n) = \widehat{p}_{ij}(t)$ with $n_{ij}(t) = n$). From Lemma II.1 of [12], this yields

$$
\mathbf{P}(i_t^* \notin \mathcal{C}_t) \leq \sum_{j = i_t^* + 1}^{K} \sum_{n=1}^{t} n \exp\Big( -\frac{\alpha \log t}{K} \Big) \leq K t^2 \exp\Big( -\frac{\alpha \log t}{K} \Big) \leq \frac{K}{t^2},
$$

since $\alpha \geq 4K$. Therefore,

$$
\sum_{t=1}^{T} \mathbf{P}(i_t^* \notin \mathcal{C}_t) \leq K \sum_{t=1}^{T} \frac{1}{t^2} \leq 2K. \tag{7}
$$

Then,

$$
\mathbf{E}\Big[ \sum_{t=1}^{T} \mathbf{1}(\mathcal{E}_t) \mathbf{1}\big(\mathcal{G}^c(t)\big) \Big] \leq \mathbf{E}\Big[ \sum_{t=1}^{T} \mathbf{1}(\mathcal{E}_t) \mathbf{1}\big(\mathcal{G}^c(t)\big) \mathbf{1}(i_t^* \in \mathcal{C}_t) \Big] + \mathbf{E}\Big[ \sum_{t=1}^{T} \mathbf{1}(i_t^* \notin \mathcal{C}_t) \Big]
$$

$$
\overset{(7)}{\leq} \mathbf{E}\Big[ \sum_{t=1}^{T} \mathbf{1}(\mathcal{E}_t) \mathbf{1}\big(\mathcal{G}^c(t)\big) \mathbf{1}(i_t^* \in \mathcal{C}_t) \Big] + 2K
$$

$$
= \mathbf{E}\Big[ \sum_{t=1}^{T} \sum_{i=1}^{K} \mathbf{1}(\mathcal{E}_t) \mathbf{1}\big(\mathcal{G}^c(t)\big) \mathbf{1}(i_t^* = i) \mathbf{1}(i \in \mathcal{C}_t) \Big] + 2K.
$$

Now, since $x_t$ is uniformly sampled from $\mathcal{C}_t$ from Line 11 of Algorithm 2, given that $i \in \mathcal{C}_t$, the probability that $x_t = i$ is at least $1/|\mathcal{C}_t| \geq 1/K$. Thus,

$$
\mathbf{E}\big[ \mathbf{1}(\mathcal{E}_t) \mathbf{1}\big(\mathcal{G}^c(t)\big) \mathbf{1}(i_t^* = i) \mathbf{1}(i \in \mathcal{C}_t) \big] \leq K \mathbf{E}\big[ \mathbf{1}(\mathcal{E}_t) \mathbf{1}\big(\mathcal{G}^c(t)\big) \mathbf{1}(i \in \mathcal{C}_t) \mathbf{1}(i_t^* = x_t = i) \big],
$$

which yields

$$
\mathbf{E}\Big[ \sum_{t=1}^{T} \mathbf{1}(\mathcal{E}_t) \mathbf{1}\big(\mathcal{G}^c(t)\big) \Big] \leq K \sum_{i=1}^{K} \mathbf{E}\Big[ \sum_{t=1}^{T} \mathbf{1}(\mathcal{E}_t) \mathbf{1}\big(\mathcal{G}^c(t)\big) \mathbf{1}(i \in \mathcal{C}_t) \mathbf{1}(i_t^* = x_t = i) \Big] + 2K
$$

$$
\overset{(*)}{=} K \sum_{i=1}^{K} \sum_{j: \Delta(i,j) > \varepsilon_i} \mathbf{E}\Big[ \sum_{t=1}^{T} \mathbf{1}(\mathcal{E}_t) \mathbf{1}(i \in \mathcal{C}_t) \mathbf{1}(i_t^* = i) \mathbf{1}\big((x_t, y_t) = (i, j)\big) \mathbf{1}(\widehat{p}_{ij}(t) < 1/2) \Big] + 2K
$$

$$
\leq K \sum_{i=1}^{K} \sum_{j: \Delta(i,j) > \varepsilon_i} \mathbf{E}\Big[ \sum_{t=1}^{T} \mathbf{1}(\mathcal{E}_t) \mathbf{1}\big((x_t, y_t) = (i, j)\big) \mathbf{1}(\widehat{p}_{ij}(t) < 1/2) \Big] + 2K \tag{8}
$$

where $(*)$ is because $\mathcal{G}^c(t) := \big\{ \exists j > i, \Delta(i, j) > \varepsilon_{i^*}, \widehat{p}_{ij}(t) < 1/2 \big\}$ when $i = i_t^*$, and since $y_t$ is chosen such that $\widehat{p}_{iy_t}(t) < 1/2$ (see Line 12. of Alg. 2). Recall that $\mathcal{E}_t$ ensures that $(i, j)$ was pulled at least $t_0$ times during the exploration phase. Recalling that $\tilde{p}_{ij}(n)$ equals $\widehat{p}_{ij}(t)$ where $t$ is such that $n = n_{ij}(t)$, we have

$$
\sum_{t=1}^{T} \mathbf{1}(\mathcal{E}_t) \mathbf{1}\big((x_t, y_t) = (i, j)\big) \mathbf{1}(\widehat{p}_{ij}(t) < 1/2) \leq \sum_{n=t_0}^{\infty} \mathbf{1}(\tilde{p}_{ij}(n) < 1/2).
$$

Therefore, plugging the latter inequality into the previous upper-bound (8), it yields

$$\mathbf{E}\left[\sum_{t=1}^{T}\mathbf{1}(\mathcal{E}_t)\mathbf{1}\big(\mathcal{G}^c(t)\big)\right] \leq K\sum_{i=1}^{K}\sum_{j:\Delta(i,j)>\varepsilon_i}\sum_{n=t_0}^{\infty}\mathbf{P}\big(\tilde{p}_{ij}(n)<1/2\big)+2K$$

$$\overset{(a)}{=} K\sum_{i=1}^{K}\sum_{j:\Delta(i,j)>\varepsilon_i}\sum_{n=t_0}^{\infty}\mathbf{P}\big(\tilde{p}_{ij}(n)<P(i,j)-\Delta(i,j)\big)+2K$$

$$\overset{(b)}{\leq} K\sum_{i=1}^{K}\sum_{j:\Delta(i,j)>\varepsilon_i}\sum_{n=t_0}^{\infty}\exp\big(-n\Delta(i,j)^2\big)+2K$$

$$\leq K\sum_{i=1}^{K}\sum_{j:\Delta(i,j)>\varepsilon_i}\frac{e^{-(t_0-1)\Delta(i,j)^2}}{e^{\Delta(i,j)^2}-1}+2K$$

$$\leq K\sum_{i=1}^{K}\sum_{j:\Delta(i,j)>\varepsilon_i}\frac{e^{-(t_0-1)\Delta(i,j)^2}}{\Delta(i,j)^2}+2K$$

where $(a)$ follows by definition $\Delta(i,j):=\mathbf{P}(i,j)-1/2$ and $(b)$ is by Hoeffding's inequality. $\qquad\square$

The high-level idea of Lemma 9 below is that for any pair $1\leq i<j\leq K$, $j$ will not be played too much more than $M_{ij}(\delta)$ times together with items $k\leq i$. In other words, after sufficiently enough rounds $j$ is detected as worse than all items $k<i$.

**Lemma 9.** *Let $1\leq i<j\leq K$. Then, SlDB-ED (Alg. 2) satisfies:*

$$\mathbf{E}\left[\sum_{t=1}^{T}\mathbf{1}(\mathcal{G}(t))\sum_{k=1}^{i}\mathbf{1}(x_t=j,y_t=k)\mathbf{1}\Big(N_{ij}(t)>M_{ij}(\delta)\Big)\right]\leq\frac{32}{\delta^2\Delta(i,j)^2},$$

*where $\mathcal{G}(t)$ is as defined in Lem. 8 and $N_{ij}(t):=\sum_{k=1}^{i}n_{kj}(t)$ is the number of times $j$ was compared with some arm in $1,\dots,i$ and $M_{ij}(\delta):=(\alpha+\delta)(\log T)/\mathrm{kl}(\mathbf{p}_{ji},0.5)$.*

*Proof.* Let $1\leq i\leq K-1$. We start by recalling some useful notations:

$$\widehat{\mathcal{B}}_i(t):=\Big\{j\mid j\in[K],\widehat{p}_{i,j}(t)\leq 1/2\Big\},\qquad \mathcal{I}_i(t):=\sum_{j\in\widehat{\mathcal{B}}_i(t)}n_{ij}(t)\,\mathrm{kl}(\widehat{p}_{ij}(t),0.5)\,,$$

where $\widehat{i}^*(t):=\arg\min_{i\in[K]}\mathcal{I}_i(t)$, and for simplicity we here denote $\mathcal{I}_{\widehat{i}_t^*}(t)=\mathcal{I}^*(t):=\min_{i\in[K]}\mathcal{I}_i(t)$. We also denote that event $\mathcal{J}_i(t):=\{\mathcal{I}_i(t)-\mathcal{I}^*(t)\leq\alpha\log t\}$. Then for any fixed $j>i$, we have

$$S_T(i,j):=\mathbf{E}\left[\sum_{t=1}^{T}\mathbf{1}(\mathcal{G}(t))\sum_{k=1}^{i}\mathbf{1}(x_t=j,y_t=k)\mathbf{1}\Big(N_{ij}(t)>M_{ij}(\delta)\Big)\right]$$

$$=\mathbf{E}\left[\sum_{t=1}^{T}\mathbf{1}(\mathcal{G}(t))\sum_{k=1}^{i}\mathbf{1}(x_t=j,y_t=k)\mathbf{1}\Big(N_{ij}(t)>M_{ij}(\delta)\Big)\mathbf{1}(\mathcal{J}_j(t))\right]\quad\leftarrow\text{ as }x_t=j\text{ implies }\mathbf{1}(\mathcal{J}_j(t))=1$$

$$=\mathbf{E}\left[\sum_{t=1}^{T}\mathbf{1}(\mathcal{G}(t))\sum_{k=1}^{i}\mathbf{1}(x_t=j,y_t=k)\mathbf{1}\Big(N_{ij}(t)>M_{ij}(\delta),\mathcal{J}_j(t)\Big)\right]$$

Substituting $\mathcal{J}_i(t):=\{\mathcal{I}_i(t)-\mathcal{I}^*(t)\leq\alpha\log t\}$, and using that $\mathcal{G}(t)$ implies $\mathcal{I}^*(t)=0$, we get

$$S_T(i,j)\leq\mathbf{E}\left[\sum_{t=1}^{T}\sum_{k=1}^{i}\mathbf{1}(x_t=j,y_t=k)\mathbf{1}\Big(N_{ij}(t)>M_{ij}(\delta),\mathcal{I}_j(t)\leq\alpha\log t\Big)\right]$$

$$\leq\mathbf{E}\left[\sum_{t=1}^{T}\sum_{k=1}^{i}\mathbf{1}(x_t=j,y_t=k)\mathbf{1}\Big(N_{ij}(t)>M_{ij}(\delta),\sum_{k\in\widehat{\mathcal{B}}_j(t)}n_{jk}(t)\,\mathrm{kl}(\widehat{p}_{jk}(t),0.5)\leq\alpha\log T\Big)\right]$$

$$\leq \mathbf{E}\left[\sum_{t=1}^{T}\sum_{k=1}^{i}\mathbf{1}(x_t = j, y_t = k)\mathbf{1}\left(N_{ij}(t) > M_{ij}(\delta), \sum_{k=1}^{i} n_{kj}(t)\,\mathrm{kl}^+(\widehat{p}_{jk}(t), 0.5) \leq \alpha \log T\right)\right]$$

where $\mathrm{kl}^+(p,q) := \mathrm{kl}(p,q)\mathbf{1}(p < q)$. But, from convexity of $\mathrm{kl}^+(\cdot, 0.5)$ together with Jensen's inequality

$$\sum_{k=1}^{i} n_{kj}(t)\,\mathrm{kl}^+(\widehat{p}_{jk}(t), 0.5) \geq N_{ij}(t)\,\mathrm{kl}^+\left(\frac{1}{N_{ij}(t)}\sum_{k=1}^{i} n_{kj}(t)\widehat{p}_{jk}(t), 0.5\right).$$

Therefore, denoting

$$\tilde{p}_{1:ij}(N_{ij}(t)) := \frac{1}{N_{ij}(t)}\sum_{k=1}^{i} n_{kj}(t)\widehat{p}_{jk}(t) = \frac{1}{N_{ij}(t)}\sum_{k=1}^{i} w_{ki}(t)$$

the frequentist empirical estimate obtained after $N_{ij}(t)$ comparisons of $j$ with any item better than $i$, we have

$$S_T(i,j) \leq \mathbf{E}\left[\sum_{t=1}^{T}\sum_{k=1}^{i}\mathbf{1}(x_t = j, y_t = k)\mathbf{1}\left(N_{ij}(t) > M_{ij}(\delta),\ N_{ij}(t)\,\mathrm{kl}^+\left(\tilde{p}_{1:ij}(N_{ij}(t)), 0.5\right) \leq \alpha \log T\right)\right]$$

But, for each $n > M_{ij}(\delta)$, $N_{ij}(t) = n$ is only possible for one of the above rounds since $(x_t, y_t) = (i, k)$ with $k \leq i$, which increases $N_{ij}(t)$ by one. Thus,

$$S_T(i,j) \leq \mathbf{E}\left[\sum_{n=M_{ij}(\delta)}^{T}\mathbf{1}\left(n\,\mathrm{kl}^+\left(\tilde{p}_{1:ij}(n), 0.5\right) \leq \alpha \log T\right)\right]$$

$$\leq \mathbf{E}\left[\sum_{n=\lceil M_{ij}(\delta)\rceil}^{T}\mathbf{1}\left(M_{ij}(\delta)\,\mathrm{kl}^+\left(\tilde{p}_{1:ij}(n), 0.5\right) \leq \alpha \log T\right)\right]$$

$$\leq \mathbf{E}\left[\sum_{n=\lceil M_{ij}(\delta)\rceil}^{T}\mathbf{1}\left(\mathrm{kl}^+\left(\tilde{p}_{1:ij}(n), 0.5\right) \leq \frac{\mathrm{kl}(p_{ji}, 0.5)}{1+\delta}\right)\right]$$

Now, let $\mu_i \in (p_{ji}, 0.5)$ such that $\mathrm{kl}(\mu_i, 0.5) = \mathrm{kl}(p_{ji}, 0.5)/(1+\delta)$. By monotonicity of $\mathrm{kl}^+(\cdot, 0.5)$,

$$S_T(i,j) = \sum_{n=\lceil M_{ij}(\delta)\rceil}^{T}\mathbf{P}\left(\mathrm{kl}^+\left(\tilde{p}_{1:ij}(n), 0.5\right) \leq \mathrm{kl}^+(\mu_i, 0.5)\right)$$

$$\leq \sum_{n=\lceil M_{ij}(\delta)\rceil}^{T}\mathbf{P}\left(\tilde{p}_{1:ij}(n) \leq \mu_i\right)$$

$$\leq \sum_{n=\lceil M_{ij}(\delta)\rceil}^{T}\mathbf{P}\left(\tilde{p}_{ij}(n) \leq \mu_i\right)$$

$$\leq \sum_{n=\lceil M_{ij}(\delta)\rceil}^{T} e^{-\mathrm{kl}(\mu_i, p_{ij})n},$$

where the last inequality is by Chernoff's inequality (e.g. see Fact 8 of [21]). Then,

$$S_T(i,j) \leq \sum_{n=1}^{\infty} e^{-\mathrm{kl}(\mu_i, p_{ij})n} \leq \frac{1}{\mathrm{kl}(\mu_i, p_{ij})}.$$

The proof is concluded using Pinksker's inequality followed by $4\Delta(i,j)$-Lipschitzness of $\mathrm{kl}(\cdot, 0.5)$ over $(0.5 - \Delta(i,j), 0.5)$:

$$\mathrm{kl}(\mu_i, p_{ij}) \geq 2(\mu_i - p_{ij})^2 \qquad\qquad \leftarrow \text{Pinsker's inequality}$$

$$\geq \frac{2}{16\Delta(i,j)^2}\left(\mathrm{kl}(\mu_i, 0.5) - \mathrm{kl}(p_{ij}, 0.5)\right)^2 \qquad \leftarrow \text{Lipschitzness}$$

$$= \frac{2\,\mathrm{kl}(p_{ij}, 0.5)^2 \delta^2}{16\Delta(i,j)^2(1+\delta)^2} \qquad\qquad \leftarrow \text{def of } \mu_i$$

$$\geq \frac{\mathrm{kl}(p_{ij}, 0.5)^2 \delta^2}{32\Delta(i,j)^2} \qquad\qquad \leftarrow \delta \in (0,1)$$

$$\geq \frac{\Delta(i,j)^2 \delta^2}{32} \, . \qquad\qquad \leftarrow \text{Pinsker's inequality}$$

Therefore,

$$S_T(i,j) \leq \frac{32}{\Delta(i,j)^2 \delta^2} \, .$$

$\square$

**Lemma 10.** *For any $\varepsilon_2, \ldots, \varepsilon_K \geq 0$,*

$$\sum_{1 \leq i < j \leq K | \Delta(i,j) > \varepsilon_j} \frac{\Delta(i,j) - \Delta(i+1,j)}{\Delta(i,j)^2} \leq \sum_{j=2}^{n} \frac{2}{\max\left\{\varepsilon_j, \Delta(j-1,j)\right\}} \, .$$

*Proof.* The proof is adapted from similar techniques used for proving Lem. 5 of [19]. First note that

$$\sum_{1 \leq i < j \leq n | \Delta(i,j) > \varepsilon_j} \frac{\Delta(i,j) - \Delta(i+1,j)}{\Delta(i,j)^2} = \sum_{i=1}^{K-1} \sum_{j \in [K]\setminus[i] | \Delta(i,j) > \varepsilon_j} \frac{\Delta(i,j) - \Delta(i+1,j)}{\Delta(i,j)^2}$$

Let us fix any arm $i \in [K-1]$, and denote by $\nabla_{i,j} := \Delta(i,j) - \Delta(i+1,j)$. Then we note

$$\sum_{j \in [K]\setminus[i] | \Delta(i,j) > \varepsilon_j} \frac{\Delta(i,j) - \Delta(i+1,j)}{\Delta(i,j)^2} = \sum_{j \in [K]\setminus[i] | \Delta(i,j) > \varepsilon_j} \nabla_{i,j} \int_0^\infty \mathbf{1}(\Delta(i,j)^{-2} \geq x) dx$$

$$= \sum_{j=i+1}^{K} \mathbf{1}(\Delta(i,j) > \varepsilon_j) \nabla_{i,j} \int_0^\infty \mathbf{1}(\Delta(i,j)^{-2} \geq x) dx$$

$$= \sum_{j=i+1}^{K} \nabla_{i,j} \int_0^\infty \mathbf{1}(\Delta(i,j) > \varepsilon_j, \Delta(i,j)^{-2} \geq x) dx$$

$$= 2 \sum_{j=i+1}^{K} \nabla_{i,j} \int_0^\infty y^{-3} \mathbf{1}(\varepsilon_j < \Delta(i,j) < y) dy \quad \leftarrow \text{ change of variable } x = y^{-2}, dx = -2y^{-3}dy$$

$$= 2 \sum_{j=i+1}^{K} \nabla_{i,j} \int_{\varepsilon_j}^\infty y^{-3} \mathbf{1}(\varepsilon_j < \Delta(i,j) < y) dy$$

Further summing over all $i \in [K-1]$, we get

$$A_T := \sum_{1 \leq i < j \leq n | \Delta(i,j) > \varepsilon_j} \frac{\Delta(i,j) - \Delta(i+1,j)}{\Delta(i,j)^2}$$

$$= \sum_{i=1}^{K-1} \left( 2 \sum_{j=i+1}^{K} \nabla_{i,j} \int_{\varepsilon_j}^\infty y^{-3} \mathbf{1}(\varepsilon_j < \Delta(i,j) < y) dy \right)$$

$$= 2 \sum_{i=1}^{K-1} \sum_{j=i+1}^{K} \int_{\varepsilon_j}^\infty \left( \nabla_{i,j} y^{-3} \mathbf{1}(\varepsilon_j < \Delta(i,j) < y) dy \right)$$

$$= 2 \sum_{j=2}^{K} \sum_{i=1}^{j-1} \int_{\varepsilon_j}^\infty y^{-3} \Big( \big(\Delta(i,j) - \Delta(i+1,j)\big) \mathbf{1}(\varepsilon_j < \Delta(i,j) \leq y) \Big) dy$$

$$= 2 \sum_{j=2}^{K} \int_{\varepsilon_j}^\infty y^{-3} \sum_{i=i_y(j)}^{i_{\varepsilon_j}(j)-1} \Big( \Delta(i,j) - \Delta(i+1,j) \Big) dy \, ,$$

where $i_\varepsilon(j) := \arg\min\{i | i \leq j, \Delta(i,j) \leq \varepsilon\}$ (with the convention that the sum is empty if the $\arg\min$ is empty) and because $\varepsilon_j < \Delta(i,j) \leq y$ is equivalent to $i_y(\varepsilon) \leq i \leq i_{\varepsilon_j} - 1$. Using telescoping summation over $i$, we further get:

$$A_T \leq 2\sum_{j=2}^{K} \int_{\varepsilon_j}^{\infty} y^{-3}\big(\Delta(i_y(j),j) - \Delta(i_{\varepsilon_j}(j),j)\big)dy$$

$$\leq 2\sum_{j=2}^{K} \int_{\varepsilon_j}^{\infty} y^{-3}\Delta(i_y(j),j)dy \quad \leftarrow \text{ since } \Delta(i_{\varepsilon_j}(j),j) > 0$$

Then, since $\Delta(i_y(j),j) = 0$ if $y < \Delta(j-1,j)$, we have

$$A_T \leq 2\sum_{j=2}^{K} \int_{\max\{\varepsilon_j,\Delta(j-1,j)\}}^{\infty} y^{-3}\Delta(i_y(j),j)dy$$

$$\leq 2\sum_{j=2}^{K} \int_{\max\{\varepsilon_j,\Delta(j-1,j)\}}^{\infty} y^{-2}dy \qquad \leftarrow \text{ since } \Delta(i_y(j),j) \leq y$$

$$\leq 2\sum_{j=2}^{K} \frac{1}{\max\{\varepsilon_j,\Delta(j-1,j)\}},$$

which concludes the proof. $\qquad\qquad\qquad\qquad\qquad\qquad\qquad\qquad\qquad\qquad\qquad\qquad\square$

## 10.2 Proof of Theorem 6

**Theorem 6** (Expected regret analysis S1DB-ED). *Let $t_0 = 1$ and $\alpha = 4K$. Then as $T \to \infty$, the expected regret incurred by SlDB-ED (Alg. 2) can be upper bounded as: For all $\varepsilon_2,\ldots,\varepsilon_K \geq 0$*

$$\mathbf{E}\big[R_T\big] \lesssim K^2 + \sum_{1\leq i<j\leq K}\left(\frac{K\mathbf{1}_{\{\Delta(i,j)>\varepsilon_j\}}}{\Delta(i,j)^2} + n_{ij}(T)\min\{\varepsilon_j,\Delta(i,j)\}\right) + \sum_{j=2}^{K}\frac{K\log T}{\max\big\{\varepsilon_j,\Delta(j-1,j)_+\big\}}$$

$$\leq O\left(\min\left\{\sum_{j=2}^{K}\frac{K\log T}{\Delta(j-1,j)_+},\ KT^{2/3}\right\}\right).$$

*Proof.* We analyse the expected regret SlDB-ED (Alg 2) for some fixed sequence $\mathcal{S}_T$. Recall that $t_0$ is the budget spent on exploration of each pair $(i,j)$ and the notation

$$\mathcal{E}(t) := \big\{n_{ij}(t) > t_0, \forall i,j \in S_t, i \neq j\big\},$$

to be the event when all distinct pairs in $S_t$ have been explored $t_0$ times and

$$\mathcal{G}(t) := \big\{\forall j > i_t^*, \Delta(i_t^*,j) > \varepsilon_i, \widehat{p}_{i_t^*j}(t) > 1/2\big\}$$

the event when the probabilities $p_{ij}$ have been well estimated by the algorithm. Then, from Lemma 8, we have

$$\mathbf{E}\big[R_T\big] = \mathbf{E}\left[\sum_{t=1}^{T} r_t\right] = \mathbf{E}\left[\sum_{t=1}^{T}\mathbf{1}(\mathcal{E}^c(t))r_t + \sum_{t=1}^{T}\mathbf{1}(\mathcal{E}(t))\mathbf{1}(\mathcal{G}^c(t))r_t + \sum_{t=T_0+1}^{T}\mathbf{1}(\mathcal{E}(t))\mathbf{1}(\mathcal{G}(t))r_t\right]$$

$$\leq K^2 t_0 + 2K + K\sum_{i=1}^{K-1}\sum_{j=i+1|\Delta(i,j)>\varepsilon_j}^{K}\frac{e^{-(t_0-1)\Delta(i,j)^2}}{\Delta(i,j)^2} + \mathbf{E}\left[\underbrace{\sum_{t=T_0+1}^{T}\mathbf{1}(\mathcal{E}(t))\mathbf{1}(\mathcal{G}(t))r_t}_{E_T}\right]. \quad (9)$$

We now upper-bound the third term of (9). Remark that under $\mathcal{G}(t)$ the algorithm chooses $y_t = i_t^*$. Therefore,

$$E_T := \sum_{t=1}^{T}\mathbf{1}(\mathcal{E}(t))\mathbf{1}(\mathcal{G}(t))r_t$$

$$\leq \sum_{t=1}^{T} \mathbf{1}(\mathcal{G}(t)) r_t$$

$$= \sum_{t=1}^{T} \mathbf{1}(\mathcal{G}(t)) \sum_{1 \leq i < j \leq K} \mathbf{1}(x_t = j, y_t = i) r_t$$

$$= \sum_{t=1}^{T} \mathbf{1}(\mathcal{G}(t)) \sum_{1 \leq i < j \leq K} \mathbf{1}(x_t = j, y_t = i) \frac{\Delta(i,j)}{2} \quad \leftarrow \text{ because } \mathcal{G}(t) \text{ implies } y_t = i_t^*$$

$$\leq \sum_{1 \leq i < j \leq K : \Delta(i,j) < \varepsilon_j} n_{ij}(T) \frac{\Delta(i,j)}{2} + \underbrace{\sum_{t=1}^{T} \mathbf{1}(\mathcal{G}(t)) \sum_{1 \leq i < j \leq K : \Delta(i,j) > \varepsilon_j} \mathbf{1}(x_t = j, y_t = i) \frac{\Delta(i,j)}{2}}_{=:D_T}$$

$$(10)$$

Moreover, recalling the notations $n_{ij}(T) := \sum_{t=1}^{T} \mathbf{1}\big(\{x_t, y_t\} = \{i,j\}\big)$ and defining

$$\tilde{N}_{ij}(T) := \sum_{k=1}^{i} \sum_{s=1}^{t} \mathbf{1}(\mathcal{G}(s)) \mathbf{1}(x_t = k, y_t = j) \leq N_{ij}(T) := \sum_{k=1}^{i} n_{kj}(T) \,,$$

we have

$$D_T := \sum_{1 \leq i < j \leq K : \Delta(i,j) > \varepsilon_j} \sum_{t=1}^{T} \mathbf{1}(\mathcal{G}(t)) \mathbf{1}(x_t = j, y_t = i) \frac{\Delta(i,j)}{2}$$

$$= \sum_{1 \leq i < j \leq K : \Delta(i,j) > \varepsilon_j} \big(\tilde{N}_{ij}(T) - \tilde{N}_{(i-1)j}(T)\big) \frac{\Delta(i,j)}{2}$$

$$= \sum_{j=2}^{K} \sum_{i=1}^{i_{\varepsilon_j}(j)} \big(\tilde{N}_{ij}(T) - \tilde{N}_{(i-1)j}(T)\big) \frac{\Delta(i,j)}{2}$$

$$= \sum_{j=2}^{K} \tilde{N}_{i_{\varepsilon_j}j}(T) \frac{\varepsilon_j}{2} + \sum_{j=2}^{K} \sum_{i=1}^{i_{\varepsilon_j}(j)-1} \tilde{N}_{ij}(T) \frac{\Delta(i,j) - \Delta(i+1,j)}{2}$$

$$\leq \sum_{1 \leq i < j \leq K : \Delta(i,j) \geq \varepsilon_j} n_{ij}(T) \frac{\varepsilon_j}{2} + \sum_{1 \leq i < j \leq K : \Delta(i,j) > \varepsilon_j} \tilde{N}_{ij}(T) \frac{\Delta(i,j) - \Delta(i+1,j)}{2} \,. \quad (11)$$

Now, we need to upper-bound $\tilde{N}_{ij}(T)$. We have,

$$\tilde{N}_{i,j}(T) := \sum_{t=1}^{T} \mathbf{1}(\mathcal{G}(t)) \sum_{k=1}^{i} \mathbf{1}\big(x_t = j, y_t = k\big)$$

$$\leq \sum_{t=1}^{T} \mathbf{1}(\mathcal{G}(t)) \sum_{k=1}^{i} \mathbf{1}\big(x_t = j, y_t = k\big) \Big[ \mathbf{1}\big(N_{ij}(t) \leq M_{ij}(\delta)\big) + \mathbf{1}\big(N_{ij}(t) > M_{ij}(\delta)\big) \Big]$$

$$\leq M_{ij}(\delta) + \sum_{t=1}^{T} \mathbf{1}(\mathcal{G}(t)) \sum_{k=1}^{i} \mathbf{1}\big(x_t = j, y_t = k\big) \mathbf{1}\big(N_{ij}(t) > M_{ij}(\delta)\big)$$

$$\leq M_{ij}(\delta) + \frac{32}{\delta^2 \Delta(i,j)^2} \quad \leftarrow \text{Lemma 9}$$

$$= \frac{(\alpha + \delta) \log T}{\mathrm{kl}(p_{ji}, 0.5)} + \frac{32}{\delta^2 \Delta(i,j)^2}$$

$$\leq \Big( 2(\alpha + \delta) \log T + \frac{32}{\delta^2} \Big) \frac{1}{\Delta(i,j)^2} \,,$$

where the last inequality comes from Pinksker's inequality. This entails

$$\sum_{1 \leq i < j \leq K : \Delta(i,j) > \varepsilon_j} \tilde{N}_{ij}(T) \frac{\Delta(i,j) - \Delta(i+1,j)}{2}$$

$$\leq \left( (\alpha + \delta) \log T + \frac{16}{\delta^2} \right) \sum_{1 \leq i < j \leq K : \Delta(i,j) > \varepsilon_j} \frac{\Delta(i,j) - \Delta(i+1,j)}{\Delta(i,j)^2}$$

$$\leq 2 \sum_{j=2}^{K} \frac{(\alpha + \delta) \log T + 16\delta^{-2}}{\max\left\{ \varepsilon_j, \Delta(j-1,j) \right\}} \, .$$

Combining this inequality with (9), (10), and (11) and choosing $t_0 = 1$ and $\alpha = 4K$ concludes

$$\mathbf{E}\big[R_T\big] \leq K^2 t_0 + 2K + K \sum_{i=1}^{K-1} \sum_{j=i+1 | \Delta(i,j) > \varepsilon_j}^{K} \frac{e^{-(t_0-1)\Delta(i,j)^2}}{\Delta(i,j)^2}$$

$$+ \sum_{1 \leq i < j \leq K} n_{ij}(T) \frac{\min\{\varepsilon_j, \Delta(i,j)\}}{2} + 2 \sum_{j=2}^{n} \frac{(\alpha + \delta) \log T + 16\delta^{-2}}{\max\left\{ \varepsilon_j, \Delta(j-1,j) \right\}}$$

$$\leq K(K+2) + \sum_{1 \leq i < j \leq K | \Delta(i,j) > \varepsilon_j} \frac{K}{\Delta(i,j)^2}$$

$$+ \sum_{1 \leq i < j \leq K} n_{ij}(T) \frac{\min\{\varepsilon_j, \Delta(i,j)\}}{2} + 2 \sum_{j=2}^{K} \frac{(4K + \delta) \log T + 16\delta^{-2}}{\max\left\{ \varepsilon_j, \Delta(j-1,j) \right\}} \, .$$

$$\square$$

*Proof.* Recall that the proof was done for any $\varepsilon_2, \ldots, \varepsilon_K \geq 0$ that are independent of the algorithm. In particular, choosing $\varepsilon_2, \ldots, \varepsilon_K = \varepsilon$ entails that for any $\varepsilon > 0$

$$\mathbf{E}\big[R_T\big] \lesssim K^2 + \varepsilon T + \sum_{1 \leq i < j \leq K | \Delta(i,j) > \varepsilon} \frac{K}{\Delta(i,j)^2} + \sum_{j=2}^{K} \frac{K \log T}{\max\left\{ \varepsilon, \Delta(j-1,j) \right\}}$$

which yields making $\varepsilon \to 0$ the distribution-dependent asymptotic upper-bound

$$\mathbf{E}\big[R_T\big] \leq O\left( K \log(T) \sum_{j=2}^{K} \frac{\mathbf{1}\{\Delta(j-1,j) > 0\}}{\Delta(j-1,j)} \right)$$

as $T \to \infty$ and for any fix $\varepsilon \geq 0$ and choosing $\delta = 1$. Furthermore, optimizing $\varepsilon_1 = \varepsilon_2 = \cdots = \varepsilon_K = \varepsilon = 2^{1/3} K T^{-1/3}$ yields the distribution-free upper-bound

$$\mathbf{E}\big[R_T\big] \leq K(K+2) + \frac{K^3}{\varepsilon^2} + \frac{T\varepsilon}{2} + \frac{(8K + 1)K \log T + 16K}{\varepsilon} \leq 2KT^{2/3} + O(K^2 + KT^{1/3} \log T) \, .$$

$$\square$$