# OpenReview forum: "Dueling Bandits with Adversarial Sleeping"
_NeurIPS.cc/2021/Conference — NeurIPS 2021 Poster_

### Official Review · Reviewer_ASCB · 2021-07-10

**Rating:** 6
**Confidence:** 5

**Summary:**

This paper studies a variant of the dueling bandit problem, where the availability of arms might differ for different time steps and consequently the learner might only choose pairs of arms from a subset of all existing arms. In other words, this variant is the preference-based counterpart of the sleeping bandit problem considered in the literature within the classical MAB problem with numerical rewards.
Under the assumption of a total order of arms and a possibly adversarial availability of the arms, the authors show a finite-time lower bound on the expected regret. Further, they suggest two algorithms for this learning scenario: one is based on the well-known RUCB for the classical dueling bandit problem, for which high-probability regret bounds are obtained. The other is a modification of the well-known RMED algorithm, for which a finite-time expected regret bound is shown. The main contribution of the paper is thus the transfer of the sleeping aspect to the dueling bandits and the theoretical analysis of the resulting problem.


**Ethical Concerns:**

There are no ethical issues with this paper

**Limitations And Societal Impact:**

Yes

**Main Review:**

# Significance/Novelty

This is the first work in the realm of dueling bandits considering the sleeping setting, which has gained much interest in the recent years in the classical multi-armed bandit problem. However, the transfer is rather straightforward, and the most interesting novelty of the paper is the modification of the RUCB algorithm by introducing the dominance sets as well as the different way of theoretical analysis for the modified RMED algorithm for this setting.

# Strengths

The difficulty of the considered problem is analyzed by deriving a lower bound holding for any (consistent) learning algorithm for this problem.
As mentioned above, the most interesting novelty of the paper is the modification of the RUCB algorithm by introducing the dominance sets as well as the different way of theoretical analysis for the modified RMED algorithm for this setting. The theoretical analysis for the latter algorithm seems to be sound.

# Weaknesses

It seems that the paper was written in a hurry, as there are many weird sentences due to typos, missing words, wrong grammar or suchlike. Additionally, there are also some minor niceties regarding the theoretical results (please see ‘’Minor things’’ below).
However, and this is currently (unfortunately) the reason for me to tend towards rejecting the paper, the high probability regret bound for SlDB-UCB stated in the supplementary material (differs from the statement in the main part) seems to be wrong, as the bound is independent of the time horizon T. Although Remark 6 tries to justify that this is not necessarily a contradiction to the lower bound result, I think that using techniques to translate high-probability bounds to bounds in expectation (see for instance the proof of Theorem 3.4 in [1] or the proof of Theorem 5 in [2]) will yield a bound in expectation  for SlDB-UCB which is independent of T as well, which, however, is then a contradiction. I hope that the authors can comment on this issue and correct me if I am wrong. I suspect that Lemma 6 has a flaw and a \log(T) should occur when bounding n_{i,j}.
Another (minor) weakness is that the authors do not consider in their experiments straightforward modifications (by restricting the corresponding choice mechanism with respect to the available set of arms) of other dueling bandits algorithms such as DTS [3]. This would have made the experimental study a little more exciting.

[1] Bubeck & Cesa-Bianchi (2012), “Regret Analysis of Stochastic and Nonstochastic Multi-armed Bandit Problems”

[2] Zoghi et al., (2014), “Relative Upper Confidence Bound for the K-Armed Dueling Bandit Problem”

[3] Wu & Liu, (2016), “Double Thompson Sampling for Dueling Bandits”


# Suggestions to improve readability

1. Even if I am wrong regarding my concern of the regret bound for SlDB-UCB, the authors should state a bound in expectation for sake of completeness as well.
2. The lower bound in Lemma 2 is strictly speaking not true, as there should be some constant on the right-hand side of the inequality. By the way, the regret of a an algorithm $\mathcal A$, that is, $R_T(\mathcal{A})$ is not rigorously introduced.
3. Furthermore, I didn’t understand the statement in lines 269-271, as I don’t see how the O(T^{2/3}) worst-case regret for other dueling bandits algorithms follows (from your reasoning). Perhaps you could address this in the response (and also be a bit more detailed in the paper).
4. You could be more precise regarding the additive terms in the regret bounds. For instance, the O(1) term in line 233 is O(K C(K,\delta)).
5. Finally, in section 2 you use jargon that is clear to any reader who is familiar with Dueling Bandits or preference-based learning, but not necessarily to those who are not. To be more precise, the terms “preference matrix” (lines 121-122) and “beats” (line 141) are not explained and also what it means for two arms that the preference probability is larger than 1/2 (line 124).


**Other minor things:**

line 112: indicator random(?) variable

Proof (sketch) of Theorem 1: It is a bit strange to speak of ‘’subintervals’’

Wouldn’t it be more reasonable to call SlDB-UCB rather SlDB-RUCB, as it is based on RUCB?

I think lines 12 and 13 in Algorithm 1 needs to be switched regarding the typo mentioned in lines 669-670. Also the in line 16 it should be n_{i,j}(t+1).

Statement of Theorem 3 (line 206): \alpha > 1/2 is also possible I think?

lines 223: holds good?

Statement of Theorem 8 (line 254):  Why is T tending against infinity?

Finally, it is worth mentioning once again that there are a lot of typos in the paper (not listed above), so that the paper needs a proper proofreading.

# Post Rebuttal

The authors have convinced me that the main result for SLDB-UCB is indeed correct and does not contradict the lower bound. The misunderstanding was caused by a somewhat unclear formulation of the result, which should be discussed in more detail to avoid such misunderstandings. However, the experiments are still a bit limited as explained above (e.g., it is straightforward to modify the mechanism of DTS to the sleeping setting considered) and the paper still has some minor flaws such as typos and unclear passages, but I think they can be improved quite quickly.



**Time Spent Reviewing:**

8

---

> ### Author Response · Authors · 2021-08-09
> **Response to Reviewer ASCB**
>
> Thanks for taking time in reviewing our paper. Please consider the clarifications below:
>
> Re. High probability bound for SlDB-UCB:  With respect, your understanding is wrong, there is no way to derive a T independent regret bound from the “fixed delta \in (0,1), high probability regret bound for SlDB-UCB”.
> As clarified in Rem6, the nature of the high probability regret bound (given a fixed delta) is very different from expected regret guarantees though we can derive the second from the first by using delta = O(1/T). The reason to set delta=1/T is because given any fixed delta, the expected regret is E[R_T] = (1-delta)*high-probability-regret(delta) + delta*T (Thm3). Here the “delta*T” term comes from the low-probability event when the algorithm fails to find the best (Condorcet) arm (which happens with probability delta) and goes on incurring O(1) regret for all T rounds. We want “delta*T” to be constant so set delta = 1/T.
> Now for any choice of alpha > 1/2, (say we set alpha = 1), note the first term ( i.e. (1-delta)*high-probability-regret(delta) ) contains a log(C(K,delta)) multiplicative factors which is O( log (K/delta) ) == O(log KT) when delta = 1/T. This results in an overall expected regret of E[R_T] = O(log T) finally --- more importantly, which is not independent of T and it is not possible either: If you find any way of deriving a T-independent expected regret bound for Alg 1, please let us know how(?). Since this has been the main concern, we sincerely request the reviewer to kindly evaluate the final score based on the clarification above.
>
> Re. Empirical comparisons: As correctly pointed, no there is no existing algorithm for the “sleeping dueling” problem and the existing dueling best-arm identification (e.g. Condorcet winner) algorithms will fail in the sleeping setup (variable availability sets) without any modification as they always try to pinpoint the single best arm out of the entire pool of K arms, instead of finding the “best-arm” in the available sets as required in our case (e.g. RUCB [37] precisely tries to capture the best items the set \mathcal B which in our case would keep changing with adversarial sequence of availabilities S_t and never converge to a single fixed item and lead to a suboptimal regret bound). Hence we thought adding comparisons for standard dueling routines won’t add any value, in particular as that being not the actual theoretical focus of this work. However, if advised, we would be happy to add additional experiments in the supplementary.
>
>
>
> Re. Comments of Improving Readability: ---------------------------------------
>
> Expected regret bound for SlDB-UCB: Will add the expected regret version of this algorithm in the main draft
>
> The lower bound in Lemma 2: Sure will add a constant on the right-hand side of the inequality or write it in the Omega() notation
>
> R_T(A) not defined: We disagree, it is precisely defined. Note R_T is introduced in Eqn1 and R_T(A) is well defined in Thm1 statement itself
>
> Why O(T^{2/3}) worst-case regret: Thanks for the question. Note that, any dueling algorithm with a 1/Delta^2 term in the regret bound (even if the term is independent of T) essentially leads to a worst-case (gap independent) regret of O(T^{2/3}) as the adversary can simply set Delta = \Theta(T^{-1/3}). This is because, either the learner has to spend at least 1/Delta^2 rounds to detect the best arm (Condorcet winner in the present case), or else has to go on incurring at least a regret of O(Delta) per round t, for all T rounds, leading to a total regret of at least min(1/Delta^2, Delta T). Now if the adversary chooses Delta = \Theta(T^{-1/3}), this boils down to O(T^{2/3}) worst-case regret.
> Will elaborate in the final version as advised
>
> C(K,\delta) in O(1) notation: Sure, will write KC(K,delta) instead (line233)
>
> “Jargons” like “preference matrix” and “i beats j”: The setting is clearly introduced in Sec2 and “beats” is interpretable from the context. We assume readers have basic familiarity with dueling literature and we provide several references are provided, still as advised will be happy to explain the term “preference matrix” more
>
>
> Re. Other minor things: ---------------------------------------
>
> line 112: Yes we meant indicator random variable -- we don’t see what is the problem here (?)
>
> “Subintervals” in proof (sketch) of Thm1: We believe it's appropriate as by construction of the hard instances for the available subsets, each of the KC2 sub-interval represents a particular pair of available items (1,2),...(K-1,K)
>
> Renaming SlDB-UCB: We agree our algorithm is actually based on the UCB estimates of pairwise preferences of the K items, which is a basic measure of estimate for any UCB based stochastic dueling algorithm as RUCB, but the same is also used in many earlier algorithms, e.g. RCS (Zoghi etal,14) or Ramamohan et al’16. Moreover, SlDB-UCB introduces many new ideas including the idea of dominance set, so we don’t think it's necessary to particularly of renaming it SlDB-RUCB
>
> Alg1 pseudocode: Line 12 and 13 are in the right order as D_i(t+1) is updated in line10 itself. Yes it should be n_ij(t+1) in line16, thanks much
>
> alpha in Thm3: You are correct, we can use any alpha > 1/2
>
> lines 223: Yes we meant “holds good”
>
> T \to \infty in Thm8: T \to infty in Thm8: Actually our regret for SlDB-ED is finite time, it’s not actually necessary to have T \to infty (as also follows from the proof of Thm8, Pg 26,27)
>
> Thanks for your comments on improving the readability, will incorporate them and proofread again

---

> > ### Author Response · Authors · 2021-08-19
> > **Post-rebuttal discussion**
> >
> > Dear Reviewer ASCB,
> >
> > We hope we have clarified your concerns, especially that our high probability bounds are sound and it does not lead to a T independent expected regret bound, we will be happy to clarify any other questions you may have. We would really appreciate it if you kindly reconsider improving the scores.
> >
> > Thanks
> > Authors

---

> > > ### Comment · Reviewer_ASCB · 2021-08-23
> > > **Sketch of T independent expected regret bound**
> > >
> > > Dear authors,
> > >
> > > I agree that choosing $\delta=1/T$ together with your argumentation leads to a $O(\log(T))$ term.
> > > However, I think that using the tail-integration technique as in the proof of Theorem 3.4 in [1] will yield a T independent upper bound:
> > >
> > > You show in the appendix that, for any $\delta>0,\alpha>1/2$ that
> > > $ \mathbb{P}(   R_T^{Sldb-UCB} \geq 2 \sum_{i=1}^{K-1} \sum_{j=i+1}^K M_{i,j} \log( 2 C(K,\delta) M_{i,j} )  ) \leq \delta. $
> > >
> > > Now, define $W = \frac{ R_T^{Sldb-UCB}}{ (1/2\alpha -1) }  -  2 \sum_{i=1}^{K-1} \sum_{j=i+1}^K M_{i,j} \log( 2 \frac{2(4\alpha-1)K^2}{2\alpha-1}  M_{i,j} ) .$ Then, your result is equivalent to
> > > $  \mathbb{P}(  W \geq  \log(1/\delta) ) \leq \delta.   $
> > >
> > > Using the tail-integration technique in the proof of Theorem 3.4 in [1], that is, that for any real-valued random variable X it holds that
> > > $ \mathbb{E} X \leq \int_0^1 \frac{1}{\delta} \mathbb{P}( X \geq  \log(1/\delta) )  d\delta,$ we obtain that $ \mathbb{E} W \leq 1,$ which implies $\mathbb{E}  R_T^{Sldb-UCB}  = O(1).$
> > >
> > > Please correct me, if I am wrong.
> > >
> > > [1] Bubeck & Cesa-Bianchi (2012), “Regret Analysis of Stochastic and Nonstochastic Multi-armed Bandit Problems”

---

> > > > ### Author Response · Authors · 2021-08-23
> > > > **Clarification to Reviewer ASCB (on why T-independent regret bound does not work for our high probability regret bound of SlDB-UCB with tail-integration technique)**
> > > >
> > > > Thanks for the question. Please note that there is a major gap in understanding, perhaps we could not make it clear enough last time.
> > > >
> > > > Our regret SlDB-UCB high probability bound holds good for a fixed GIVEN delta \in (0,1) (given as INPUT to the algorithm), rather than the bound to be true simultaneously for all delta \in (0,1). Please note the natures of these two results are extremely different --- the latter is a stronger guarantee where the algorithm does not require any fixed delta as the input and the high probability bound holds good SIMULTANEOUSLY for all delta \in (0,1).
> > > >
> > > > We certainly are aware of the tail integration technique used in Thm 3.4 of Bubeck & Cesa-Bianchi (2012) as pointed. In fact, the same has been used in one of the most popularly referenced dueling bandit algorithm "RUCB (Zoghi et al, 2014)" to derive expected regret bound from high probability guarantee (see Thm 4 of Zoghi et al, 2014 http://proceedings.mlr.press/v32/zoghi14.pdf, and how they used this high probability bound to derive their expected regret in Thm 5 using exactly the tail-integration technique). Now note they could use this technique only because their Thm 4 holds good SIMULTANEOUSLY for all delta \in (0,1), not a fixed delta of choice (given as input parameter). Hence their result depends polynomially on 1/delta (precisely O(1/delta)^{2alpha -1} in their C(delta) term, Thm 4) --- a much worse dependency than our "log(1/delta) term" as shown in our supplementary analysis of Thm 3. But this is expected, as otherwise, they would have ended up deriving an O(1) -- T independent bound for Thm 5 (using the same proof technique of tail-integration of the delta), which is information-theoretically impossible.
> > > >
> > > > So since our high probability bound of SlDB-UCB considers a given fixed delta \in (0,1) setting, we achieve a better O(log 1/delta) bound in terms of delta (compared to the poly(1/delta) bound of Zoghi et al / RUCB's Thm. 4), but because our result does not hold good for all delta simultaneously, we CAN NOT apply the tail integral technique. The best delta to tune for us is to choose delta = 1/T and use that as the input parameter to our algorithm which leads to the same O(log T)  bound, as we detailed earlier.
> > > >
> > > > Hope that clarifies the confusion, but will be happy to clarify if there is anything unclear still.

---

> > > > > ### Comment · Reviewer_ASCB · 2021-08-24
> > > > > **Issue resolved**
> > > > >
> > > > > Dear authors,
> > > > > thank you very much for the clarification. Now I see the difference between your results and RUCB's high probability result. However, as our discussion shows, it is worth describing your result more clearly to avoid such misunderstandings. In particular, the resulting bound for the expected regret should also be included.
> > > > >
> > > > > As a consequence, I will increase my score to 6, as the experiments are still a bit limited as explained above (e.g., it is straightforward to modify the mechanism of DTS to the sleeping setting considered) and the paper has some minor flaws such as typos and unclear passages, but I think they can be improved quite quickly.

---

> > > > > > ### Author Response · Authors · 2021-08-24
> > > > > > **Thanks**
> > > > > >
> > > > > > Dear Reviewer ASCB,
> > > > > >
> > > > > > Glad that the concern is clarified. Many thanks for confirming and re-considering the scores. We will certainly detail our regret bounds for SlDB-UCB more clearly in the update, including the dependency on delta, the high probability and the corresponding expected regret bound. Thanks for your suggestions.
> > > > > >
> > > > > > Thanks
> > > > > > Authors

---

### Official Review · Reviewer_zsxN · 2021-07-15

**Rating:** 6
**Confidence:** 3

**Summary:**

This paper studies the sleeping dueling bandits with stochastic preferences and adversarial availabilities (DB-SPAA). The paper provides a lower bound for this problem, and proposes two algorithms with near optimal regret guarantees. Experimental results are shown to demonstrate the effectiveness of the proposed algorithms.

**Limitations And Societal Impact:**

Yes

**Main Review:**

Overall, I think that the considered problem of this paper is interesting, and the theoretical analysis seems solid. However, there are still some weaknesses.
1.	The proposed S1DB-ED algorithm is too similar to RMED (Komiyama et al. 2015), so I think the novelty of this part is limited. The paper needs to give a sufficient discussion on the comparison with RMED.
2.	The comparison baselines in experiments are not sufficient. The paper only compares the proposed two algorithms, so readers cannot evaluate the empirical performance of the proposed algorithms. While I understand that this is a new problem and there are no other existing algorithms for this problem, the paper can still compare to some ablation variants of proposed algorithms to demonstrate the effectiveness of key algorithmic components, or reduce the setting to conventional dueling bandits and compare with existing dueling bandit algorithms.

**After Rebuttal**

I read the rebuttal of the authors. Now I agree that the analysis for the S1DB-ED algorithm is non-trivial and the authors correct the errors in prior work [21]. My concerns are well addressed. So I will keep my score.

**Time Spent Reviewing:**

3

---

> ### Author Response · Authors · 2021-08-09
> **Response to Reviewer zsxN**
>
> Thanks for your careful reviews and questions.
>
> Re. comparison to RMED and contributions: With respect, we strongly disagree with the comment that “... S1DB-ED algorithm is too similar to RMED[21], so I think the novelty of this part is limited...”.  Please note we already clarified the algorithmic and technical novelties of our algorithm over RMED (towards adapting to the sleeping setup) in the paper (line256-267). Moreover, we believe to have found a mistake in the original RMED’s proof due to which we had to propose new analysis for some of the key lemmas. In fact, the reviewer is requested to note that we finally ended up improving the original regret bound of the RMED algorithm in the standard K-armed dueling set up as clarified in Rem5.
> (We elaborate the details again form completeness):
> Admittedly we borrowed some ideas from RMED, but our regret analysis for SlDB-ED is highly non-trivial (and original) due to the additional `sleeping factor that it needs to take care of. Firstly, despite much effort, we could not follow the same proof analyses of RMED to find the necessary initial exploration T_{init}, after which the event of Lem 10 is almost certain to hold good. To be honest, we believe RMED's proof of Lem. 5 (equivalent of our Lem 10) contains an error where they factorize out e^{(-f(K))} in the integration (Pg 21, Appendix C, [21]) which is mathematically wrong, and correcting this already yield an exponential dependence on K (precisely f(K) = O(e^K) in their Thm. 3). So we used our very own analysis of Lem10, as can be verified just by comparing the techniques. Next, since we deal with adversarial-sleeping setup (unlike RMED), to bound the regret it is necessary to bound the number of `suboptimal arm pulls' (\neq (i_t^*,i_t^*)) as shown in Lem11---the argument was hence quite far from RMED's line of analysis for which any pair outside (1,1) is considered suboptimal irrespective of S_t. Finally, the idea of Lem 12 was completely missing in RMED since it is very much tied to the `sleeping aspect' of the problem, and precisely this is what makes our regret guarantee adaptive to the availability sequence \mathcal S_T (see Rem3). Merging all these above claims systematically led to the final regret bound of Alg2 (Thm2). Thus there are quite a few new techniques we proposed to adopt the empirical divergence idea in the sleeping dueling setup and also improved the RMED regret bound for standard Condorcet dueling regret (Rem5). We request the reviewer to kindly evaluate the final score based on these.
>
> The comparison baselines in experiments are not sufficient. The paper only compares the proposed two algorithms, so readers cannot evaluate the empirical performance of the proposed algorithms. While I understand that this is a new problem and there are no other existing algorithms for this problem, the paper can still compare to some ablation variants of proposed algorithms to demonstrate the effectiveness of key algorithmic components, or reduce the setting to conventional dueling bandits and compare with existing dueling bandit algorithms.
> Re. experiments: As we emphasized (see Sec1) and correctly mentioned by the reviewer, there is no existing algorithm for the “sleeping dueling” problem and the existing dueling best-arm identification (e.g. Condorcet winner) algorithms will fail in the sleeping setup (variable availability sets) without any modification as they always try to pinpoint the single best arm out of the entire pool of K arms, instead of finding the “best-arm” in the available sets as required in our case (e.g. RUCB [37] precisely tries to capture the best items the set \mathcal B which in our case would keep changing with adversarial sequence of availabilities S_t and never converge to a single fixed item and lead to a suboptimal regret bound). Hence we thought adding comparisons for standard dueling routines won’t add any value, in particular as that being not the actual theoretical focus of this work. However, if advised, we would be happy to add additional experiments in the supplementary.

---

> > ### Comment · Reviewer_zsxN · 2021-08-15
> > **Response to Authors - Proof of Lemma 10/Lemma 5 in [21]**
> >
> > Thank you for your reply.
> > I check the proof that the authors mentioned and compare it to [21]. It seems that the authors fix the error in the proof of Lemma 5 in [21].
> > Could you explain the idea of how to avoid the exponential factor? Why does [21] suffer such an exponential factor, and how did you handle this challenge using different techniques?
> >
> > BTW, it would be better if the authors could improve their writing/format of rebuttal, e.g., using formulas instead of latex commands. It is difficult for reviewers to quickly understand your justification by such a dense paragraph.

---

> > > ### Author Response · Authors · 2021-08-18
> > > **Clarification: Proof of Lemma 10/Lemma 5 in [21]**
> > >
> > > Thanks for the question. We bound the `probability of bad events’ (where the Condorcet arm / best-arm of the subset at time t appears empirically suboptimal, event \mathcal{G}^c_t) in our Lem 10 (an equivalent Lem 5 in Komiyama et al [21]). Now the main issue with the proof of [21] is that it is built on the proof on Honda and Takemura (2010), which requires them to decompose the bad-event [[ \mathcal{U}^c(t) in their Eqn. (19) ]] as the union of exponentially many events, one for each S \in 2^[K] subset which seems to be an overkill for the analysis. This finally leads to a union bounding over the probability of bad-events for 2^K sets, leading to exponential dependencies on K (analysis at the end of Pg. 21 in [21]). (Moreover, as mentioned in the original rebuttal, we also believe the entire proof is erroneous as there is an error earlier in their analysis in Pg 21, where they factorize out e^{(-f(K))} in the integration.)
> > > Our proof of Lem. 10 (which is fairly technical) but overall we followed a different route to bound the probability of ‘bad-event’ by decomposing it only over O(K) many arms j \in [K] for each choice of i_t^* \in [K] -- precisely we first show that, if each pair (i,j) is pulled `sufficiently enough’ (mathcal{E}_t) at round t, then whp i_t^* \in C_t (set of ‘potential condorcet arms’), see our Eqn.(4) which shows the complement event is less likely to happen. Now given i_t^* \in C_t (i.e. the best arm at time t is already in the `good-set’) whp, in the rest of the analysis we show the probability of ‘bad event’ (that the best-arm is beaten by another arm empirically) is bounded with standard algebra and careful application of Hoeffding’s concentration on \hat_p_{ij}, as detailed in Pg 20 and 21. Overall our analysis does not involve taking a union bound over an exponentially large number of terms, unlike [21], and we get an improved bound. Will be happy to clarify if any further explanation / detailing is required.
> > >
> > > P.S. Sincere apologies for the late response, we could not get back earlier due to some unavoidable personal circumstances.

---

> > > > ### Comment · Reviewer_zsxN · 2021-08-19
> > > > **Response to the Authors**
> > > >
> > > > Thank you for your clear replies. Now I understand your unique analytical techniques here. I will keep my score.

---

> > > > > ### Author Response · Authors · 2021-08-19
> > > > > **Response to Reviewer zsxN**
> > > > >
> > > > > Thanks a lot.

---

### Official Review · Reviewer_gxX7 · 2021-07-17

**Rating:** 7
**Confidence:** 3

**Summary:**

This paper studies the classic dueling bandits problem in the sleeping bandits setting, i.e. where each round an adversarially selected subset of actions is available. As in dueling bandits, each round the learner is allowed to choose two of the available actions, and learns a noisy signal about which action is better. We also assume there is a consistent total ordering of the items. The learner would like to minimize their total regret over T rounds compared to an algorithm which knows this ordering in advance.

This paper proves the following results:
- They show an Omega(K^2 log T) (really Omega(log T * sum 1/gap_{i, j})) lower bound on the regret of any learning algorithm.
- They then give an algorithm (S1DB-UCB) that matches this lower bound. The idea, roughly, is to maintain a confidence interval for each pair of items representing the probability that one beats the other. Each round, the algorithm picks one random element from the set of “potential winners” (roughly, items which have a chance of being the best item), and then a second item being the one with the highest UCB probability of beating the first item.
- They give a slightly more computationally efficient algorithm (taking ~O(K) time per step instead of O(K^2) time per step) with a slightly worse regret bound.
- Finally, they run some empirical simulations comparing the two above algorithms on synthetic data.


**Limitations And Societal Impact:**

I have no concerns here.

**Main Review:**


I think this is a pretty nice paper. The problem is very natural (I’m surprised it hasn’t been studied in the bandit setting previously), and the paper provides a pretty comprehensive set of results. Nothing strikes me as incredibly technically novel, but definitely the details of the algorithms and their analysis are non-trivial. Overall the paper was well-written, although I would prefer a clear single description somewhere of both algorithms (BB-UCB and UB-UCB) and the regret bounds they achieve; I found that I had to piece these together across multiple paragraphs of exposition.
Dueling Bandits with Adversarial Sleeping

This paper studies the classic dueling bandits problem in the sleeping bandits setting, i.e. where each round an adversarially selected subset of actions is available. As in dueling bandits, each round the learner is allowed to choose two of the available actions, and learns a noisy signal about which action is better. We also assume there is a consistent total ordering of the items. The learner would like to minimize their total regret over T rounds compared to an algorithm which knows this ordering in advance.

This paper proves the following results:
- They show an Omega(K^2 log T) (really Omega(log T * sum 1/gap_{i, j})) lower bound on the regret of any learning algorithm.
- They then give an algorithm (S1DB-UCB) that matches this lower bound. The idea, roughly, is to maintain a confidence interval for each pair of items representing the probability that one beats the other. Each round, the algorithm picks one random element from the set of “potential winners” (roughly, items which have a chance of being the best item), and then a second item being the one with the highest UCB probability of beating the first item.
- They give a slightly more computationally efficient algorithm (taking ~O(K) time per step instead of O(K^2) time per step) with a slightly worse regret bound.
- Finally, they run some empirical simulations comparing the two above algorithms on synthetic data.

The sleeping bandits setting is an interesting setting for dueling bandits. I think this paper does a thorough job of studying the dueling bandits problem in this setting (getting e.g. tight lower / upper bounds). The algorithms and their analysis are not incredibly novel, but they do appear non-trivial. In particular, I am not super familiar with the existing algorithms for dueling bandits, but some of the complexity of the analysis seems rooted in the sleeping model (as the authors point out).

The writing is in general good, although I found it a little confusing to read in some places. I think overall the paper could benefit from a slightly more in-depth summary of the previous work on dueling bandits (see question below).

Questions:

In remark 2, you mention that S1DB-UCB also improves the state of the art regret bound for regular UCB-based dueling algorithms (in particular, you say that S1DB-UCB is O(K log T / Delta) in the non-sleeping case, whereas existing state-of-the-art is O(K/Delta^2))? I’m not super familiar with the dueling bandits literature, but I imagine there is some catch here? E.g., does R-UCB also require a total ordering assumption like these algorithms? If there’s no catch, this seems like a significant result that you should emphasize more. In either case, I think it would be very useful for this paper to explain more thoroughly the prior work on dueling bandits, to understand how this work’s algorithms and model differ.

Minor Comments:

Page 9: “though the later” -> “though the latter”
Page 6: “computatinally” -> “computationally”


**Time Spent Reviewing:**

2.5

---

> ### Author Response · Authors · 2021-08-09
> **Response to Reviewer gxX7**
>
> Thanks for appreciating our work and your insightful comments.
>
> Re. SlDB-UCB vs the state of the art for the non-sleeping case: Thanks for the question. Indeed RUCB setup is more general which does not require the total ordering assumption but only the existence of a Condorcet winner arm. In particular, RUCB (Thm4, [37]) also proves a (1-delta)-high probability regret bound for any delta (unlike a given delta as assumed in our Thm3). We should have clarified the differences better and will certainly elaborate on the differences in the final version.
>
> Thanks a lot for pointing the typos, will correct them in the final version.

---

### Decision · Program_Chairs · 2021-09-27

**Decision:**

Accept (Poster)

**Comment:**

The reviewers are unanimous that this is an interesting problem. While there were some initial concerns about the analysis and relation to other algorithms, this has been cleared up in the response and discussion. Please take the reviewers minor comments into consideration when preparing the final revision. Note also the confusions of the reviewers on certain issues explained in the response will likely also be confusing to other readers, so the final revision should take special care to clarify these potential pitfalls.